# Impaired bidirectional communication between interneurons and oligodendrocyte precursor cells affects social cognitive behavior

Li-Pao Fang [1], Na Zhao [1], Laura C. Caudal [1], Hsin-Fang Chang [2], Renping Zhao [3], Ching-Hsin Lin [2], Nadine Hainz[4], Carola Meier[4], Bernhard Bettler [5], Wenhui Huang[1], Anja Scheller [1], Frank Kirchhoff [1,6 ✉] & Xianshu Bai [1 ✉]

Cortical neural circuits are complex but very precise networks of balanced excitation and inhibition. Yet, the molecular and cellular mechanisms that form the balance are just beginning to emerge. Here, using conditional γ-aminobutyric acid receptor B1- deficient mice we identify a γ-aminobutyric acid/tumor necrosis factor superfamily member 12-mediated bidirectional communication pathway between parvalbumin-positive fast spiking interneurons and oligodendrocyte precursor cells that determines the density and function of interneurons in the developing medial prefrontal cortex. Interruption of the GABAergic signaling to oligodendrocyte precursor cells results in reduced myelination and hypoactivity of interneurons, strong changes of cortical network activities and impaired social cognitive behavior. In conclusion, glial transmitter receptors are pivotal elements in finetuning distinct brain functions.

[1] Molecular Physiology, Center for Integrative Physiology and Molecular Medicine (CIPMM), University of Saarland, 66421 Homburg, Germany. [2] Cellular Neurophysiology, CIPMM, University of Saarland, 66421 Homburg, Germany. [3] Biophysics, CIPMM, University of Saarland, 66421 Homburg, Germany. [4] Department of Anatomy and Cell Biology, University of Saarland, 66421 Homburg, Germany. [5] Department of Biomedicine, University of Basel, 4056 Basel, Switzerland. [6] Experimental Research Center for Normal and Pathological Aging, University of Medicine and Pharmacy of Craiova, 200349 Craiova, Romania. ✉email: frank.kirchhoff@uks.eu; xianshu.bai@uks.eu

The inhibition of cortical network activity is performed by interneurons that release the inhibitory transmitter γ-aminobutyric acid (GABA) which acts on ionotropic $GABA_A$ and metabotropic $GABA_B$ receptors ($GABA_BR$)[1,2]. Alterations of interneuron cell density as well as concomitant changes of firing activity are often observed in several neuropsychiatric conditions[3–6]. In the mouse cortex, higher frequencies of spontaneous inhibitory postsynaptic currents (sIPSCs) were observed after increasing the interneuron density by transplantation of precursor cells[7]. Similarly, blocking of interneuron apoptosis caused a decrease of the excitation/inhibition (E/I) ratio[6,8]. Interneuron density is a pivotal determinant of correct inhibitory circuits in the central nervous system (CNS). During development, a surplus of interneurons is generated that populates the cortical plate[9]. Subsequently, about 40% of cortical inhibitory neurons are eliminated by programmed cell death during the first two postnatal weeks[7]. One hypothesis links the apoptosis of supernumerary neurons with the onset of first neuron-neuron connections[10]. Only the neurons that receive sufficient neurotrophic signals from their target cells will survive[10,11].

Oligodendrocyte precursor cells (OPCs) receive synapses from interneurons as early as postnatal day (p) 4–5[12]. Concomitantly, the incidence of interneuron apoptosis increases drastically after p5 and reaches its peak at p7[7]. Right after, the interneuron-OPC connectivity accelerates till p10, which drives an immediate oligodendrocyte (OL) boom with subsequent myelination of interneurons regulated by OPC-$GABA_A$ and $GABA_B$ receptors[13–15]. Despite relatively short axons, myelination of cortical interneurons (mainly parvalbumin (PV)$^+$ fast-spiking interneurons[16]) contributes significantly to the fine-tuning of local activity in the cortex, including the medial prefrontal cortex (mPFC), ensuring correct behavioral performance[4,17]. However, how the early communication between interneurons and OPC affects inhibitory network activity still remains unknown.

To assess the function of OPC $GABA_BRs$ for cortical inhibition, we conditionally deleted the $GABA_B$ receptor subunit gabbr1 selectively in OPCs at the end of the first postnatal week. Focusing on the mPFC, we found that OPCs shape the inhibitory network by $GABA_B$ receptors and the cytokine tumor necrosis factor-like weak inducer of apoptosis (TWEAK) signaling pathways, thereby regulating interneuron cell death and survival as well as its myelination onset. The functional and morphological changes of PV$^+$ interneurons observed in the adult mPFC of mutant mice resulted in reduced inhibitory tone, which finally caused impaired cognition and perturbed social behavior.

## Results

### $GABA_BRs$ of OPCs are required for oligodendrogenesis in the mPFC.
To assess the function of $GABA_BRs$ for OPC-interneuron communication, we generated Tg(NG2-CreER$^{T2}$):GABA$_{B1}$R$^{fl/fl}$ mice (Fig. 1a) to conditionally knockout (cKO) $GABA_BR$ from OPCs and their progeny. We induced the cKO at postnatal day 7 and 8 (p7/8) (Fig. 1a) before the onset of OL formation at p10–11[13]. When analyzed at the age of 9 weeks (w), about 76% of OPCs (platelet-derived growth factor receptor α$^+$ (PDGFRα$^+$, Pα$^+$)) were found reporter recombined in the cKO medial prefrontal cortex (mPFC), based on tdTomato (tdT) gene expression (Pα$^+$tdT$^+$/Pα$^+$, Fig. 1b, c).

To confirm and quantify the deletion of $GABA_BR$ in OPCs, we performed Western blot analysis with OPCs purified by magnetic-activated cell sorting (MACs, Supplementary Fig. 1a). In cKO OPCs, $GABA_BR$ expression was reduced by about 64.2% compared to controls (ctl) (Fig. 1d). Considering the substantial purity of MACs OPCs (85%, Supplementary Fig. 1b) and recombination efficiency (80%, Fig. 1c), we concluded that a vast majority of $GABA_BRs$ had been ablated from OPCs. In addition,

as suggested by the previous study[18], the reporter recombination efficiency (76%) faithfully indicated the extent of GABA$_{B1}$ deletion in our mice (64.2%/85% (MACs-OPC purity) = 75.5% of GABA$_{B1}$ ablation).

Next, we focused on the oligodendrocyte lineage and evaluated the contribution of $GABA_BRs$ for OPC differentiation in the mPFC by immunostaining of PDGFRα, CC1 and Olig2 (Fig. 1e), established markers for OPCs, OLs and the whole lineage, respectively. In cKO mice, the density of OPCs (PDGFRα$^+$ Olig2$^+$) did not change (Fig. 1f), while that of mature OLs (CC1$^+$Olig2$^+$) was strongly reduced by 26% (Fig. 1g, 8.3 to 6.1 cells/1 × 10$^{-3}$mm$^3$), indicative of decreased OPC differentiation to OLs (Fig. 1h). A similar reduction was observed in the primary motor cortex (MOp) (Supplementary Fig. 2a), where OPC-$GABA_BR$ deletion induced a decrease of OL density without affecting the OPC population (Supplementary Fig. 2). Notably, in the corpus callosum the densities of OLs and OPCs, as well as their relative proportions did not differ between ctl and cKO mice (Fig. 1f–h, Supplementary Fig. 3). These data suggested that OPC-$GABA_BRs$ were involved in OPC differentiation in gray matter regions such as mPFC and MOp, but not in the white matter corpus callosum.

We further analyzed structural aspects of myelination with impact on axonal conductivity, i.e., the lengths of nodes (the non-insulated gaps between paired contactin-associated protein-positive (Caspr$^+$) segments) or paranodes (single Caspr$^+$ segments indicating uncompacted myelin lamellae), by immunostaining (Fig. 1i). In cKO mPFC, the paranodal length was increased by 17% (Fig. 1j, Supplementary Fig. 4a–c) while the density and the uninsulated length of nodes were not affected (Supplementary Fig. 4d–f). In parallel, we observed a reduction of myelin basic protein (MBP) expression at mRNA (Supplementary Fig. 4g) and protein levels (Fig. 1k) in cKO mPFC, while the mRNA levels of other myelin proteins such as proteolipid protein (PLP) and myelin oligodendrocyte glycoprotein (MOG) were not changed (Supplementary Fig. 4g). Although we had observed an impaired OPC differentiation in MOp (Supplementary Fig. 2), the length and density of paranodes and nodes as well as the MBP expression were unaffected in the mutant mice (Supplementary Fig. 4h–o). Similarly, in the corpus callosum, myelination was unperturbed as indicated by unchanged paranode length and density (Supplementary Fig. 5a–e) or MBP expression (Fig. 1k). Thereby, our data showed that OPC-$GABA_BRs$ were involved in organizing nodal structures of OLs in the mPFC, but not in the MOp or corpus callosum.

Since OPCs continuously differentiate[19–21], cKO mice will also have OLs lacking $GABA_BRs$. To evaluate the cell-specific role of $GABA_BR$ and to distinguish between OPCs and mature OLs, we selectively targeted OLs using TgN(PLP-Cre$^{ERT2}$):GABA$_{B1}$R$^{fl/fl}$ mice (Supplementary Fig. 6a). The gene deletion was induced by tamoxifen injections either at p7/8 or at 4w (Supplementary Fig. 6b). At the age of 9w, we did not observe differences in the OPC and OL population between ctl and cKO mice (Supplementary Fig. 6c–f), neither in the mPFC nor in the corpus callosum. In addition, MBP expression was not altered neither in mPFC nor in corpus callosum (Supplementary Fig. 6g). These data suggested that the phenotypic changes observed in OPC-$GABA_BR$ cKO mice resulted from the functional loss of $GABA_BRs$ in OPCs rather than OLs.

In summary, our data strongly suggest that $GABA_BR$ was specifically required for early OPC differentiation and subsequent myelination in the mPFC, but not in the MOp or corpus callosum.

### Increased density of PV neurons with deteriorated myelination in cKO mPFC.
Since callosal axons are primarily excitatory[22] and

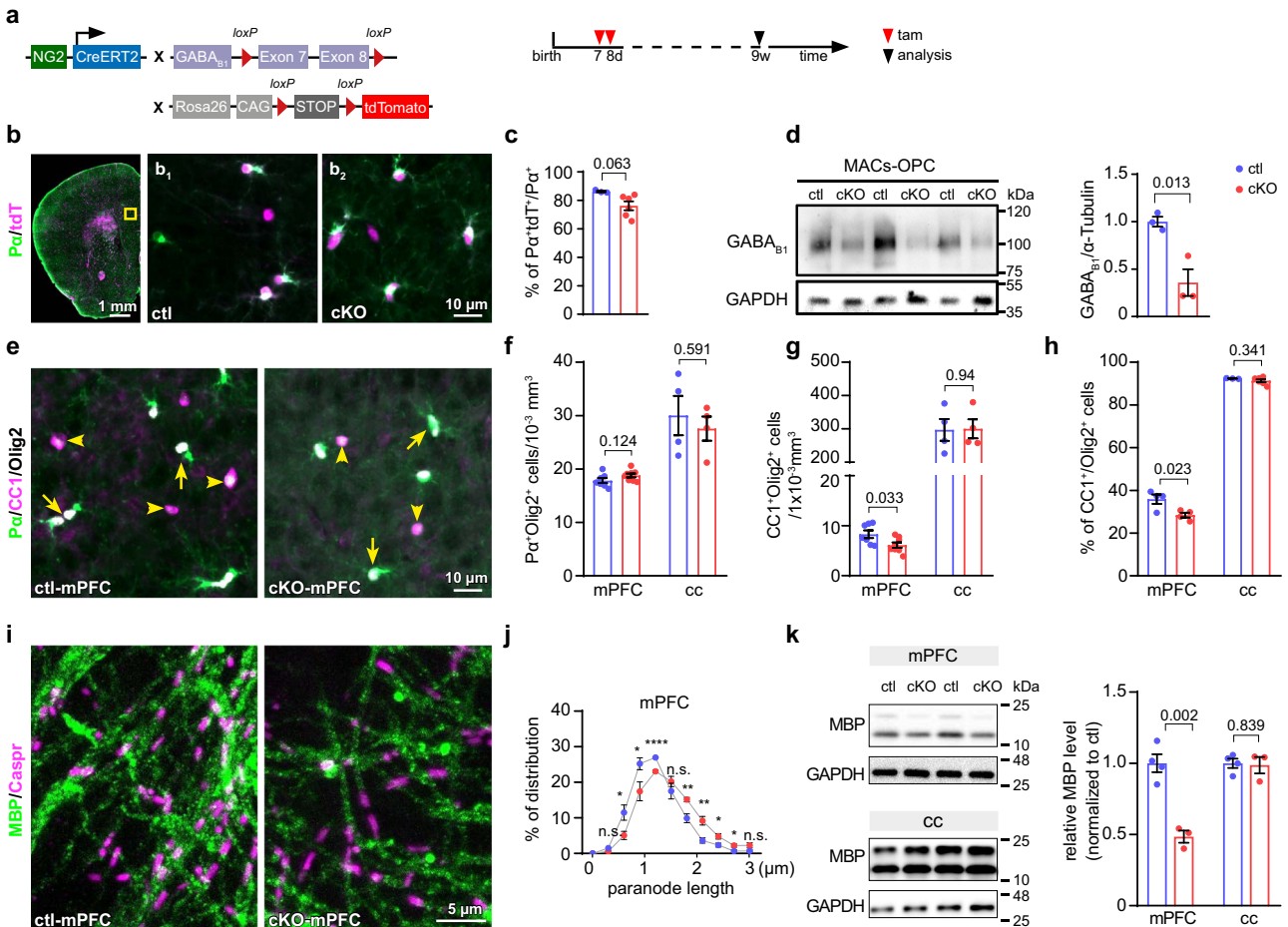

**Fig. 1 Ablation of GABA_BRs in OPCs attenuates oligodendrocyte differentiation and alters myelination in the medial prefrontal cortex. a** Mouse line and experimental schedule for OPC-specific and temporal control of GABA_BR deletion. **b** Coronal section of the prefrontal cortex immune-stained for PDGFRα (Pα, green). Expression of tdTomato$^+$ (tdT$^+$, magenta) indicates recombined cells. **b_1, b_2** Magnified micrographs of medial prefrontal cortex (mPFC, highlighted in **b**) of control (ctl) and conditional knockout (cKO) mice. **c** Recombination efficiency of OPCs in ctl and cKO mPFC. (ctl = 3 mice, cKO = 6 mice, two-sided unpaired $t$-test). **d** Western blot analysis of GABA_BR subunit 1 in magnetic-activated cell sorted (MACs) OPCs from cortex. (ctl = 3 mice, cKO = 3 mice, two-sided unpaired $t$-test). **e** Olig2 (white), Pα (green) and CC1 (magenta) immunostaining in adult ctl and cKO mPFC. **f, g** Quantification of OPC and oligodendrocyte (OL) densities in mPFC and corpus callosum (cc). (mPFC: ctl = 7 mice, cKO = 8 mice; cc: ctl = 4 mice, cKO = 4 mice, two-sided unpaired $t$-tests). **h** Quantification of the OL proportion among the total lineage (mPFC: ctl = 4 mice, cKO = 4 mice; cc: ctl = 3 mice, cKO = 6 mice, two-sided unpaired $t$-tests). **i** Immunostaining of Caspr (magenta) and MBP (green) in ctl and cKO mPFC. **j** Quantitative analysis of paranode length in mPFC (ctl = 4 mice, cKO = 4 mice, two-sided unpaired $t$-tests: $p$ = 0.193 (n.s.), 0.038 (*), 0 048 (*), 7.67E-05 (****), 0.34 (n.s.), 0.009 (**), 0.008 (**), 0.044 (*), 0.03 (*), 0.092 (n.s.), respectively). **k** Western blot analysis of MBP expression in mPFC and cc. (ctl = 4 mice, cKO = 3 mice, two-sided unpaired $t$-tests). Data are shown as mean ± SEM in **c**, **d**, **f–h**, **j** and **k**. Source data are provided as a Source Data file.

axons of inhibitory interneurons are largely restricted to cortical areas of the same hemisphere (including mPFC, Fig. 2a–c), we hypothesized that the observed myelin deficits were restricted to mPFC interneurons. Parvalbumin (PV)$^+$ interneurons, the fast-spiking interneurons, are the most abundantly myelinated interneurons of the cortex[16,23–25]. These neurons are most prominent at layers IV and V with a majority of their axons projecting to layers II/III and V[26]. To evaluate their myelination, we performed PV, neurofilament (SMI 312, pan axonal marker) and MOG triple immunostaining (Fig. 2d). We volume-rendered the MOG immunolabel as indicator of myelin sheaths and normalized the volume of MOG$^+$ fragments that covered the PV$^+$ axons (PV$^+$SMI 312$^+$) to the total volume of PV$^+$ axons at the cortical layers II/III of mPFC. Indeed, the volume of myelin sheaths wrapping PV axons was reduced by 60% in the cKO mPFC (Fig. 2e). However, the total volume of MOG immunolabel covering all SMI 312$^+$ axons was not affected (Fig. 2e), in agreement with the unchanged MOG mRNA levels in control and

mutant mice (Supplementary Fig. 4g). Also the myelination of excitatory neurons (in the corpus callosum) was not affected as indicated by an unaltered g-ratio (the ratio between the inner and the outer diameter of the myelin sheath) or unchanged conduction velocity of compound action potentials (Supplementary Fig. 5f–i). Thus, we concluded that GABA_BR of OPCs were preferentially involved in the myelination of inhibitory rather than excitatory neurons.

While investigating the interneuronal myelination, we recognized that early ablation of GABA_BR in OPCs was associated with a 25% increase of interneuron density (Fig. 2f, g, 24.3 to 30.5 cells/ $1 \times 10^{-3}$ mm$^3$) in the adult mPFC as revealed by immunostaining for PV (Fig. 2f). A detailed analysis of PV neuron density in three sub-regions of the mPFC (layer I, layer II/III and layer V/VI, Fig. 2f, g) showed a general overpopulation of PV interneurons in all cortical layers of the adult mPFC. However, the proportion of PV$^+$SMI 312$^+$ axons to all axons in layer II/III was identical between the cKO and ctl groups (Fig. 2h). These data suggested a

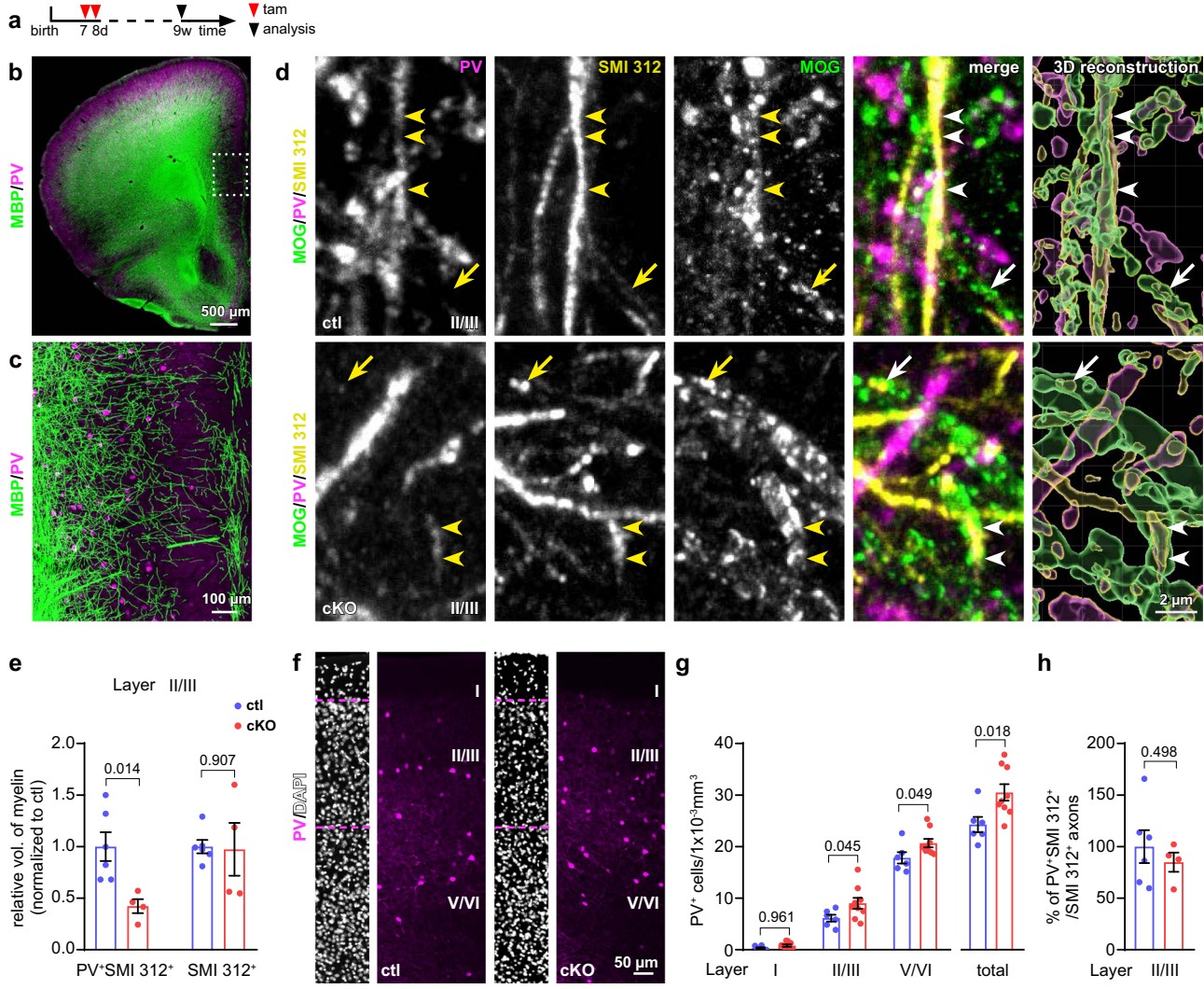

**Fig. 2 Reduced myelination and increased cell density of interneurons in OPC-GABA$_B$R cKO mice. a** Experimental schedule. **b**, **c** Overview of coronal brain slices immunostained for parvalbumin (PV, magenta) and MBP (green). PV$^+$ neurons are abundantly myelinated in the mPFC. **d** Magnified micrographs of myelinated PV axons immunostained for PV (magenta), neurofilament (with SMI 312 antibody, yellow) and myelin-oligodendrocyte glycoprotein (MOG, green) in layers II/III (II/III) ctl and cKO. The 3D reconstruction was performed using Imaris. Arrows: myelinated PV$^-$SMI 312$^+$ axons; arrowheads: myelinated PV$^+$SMI 312$^+$ axons. **e** Relative volume of MOG immunolabel around PV$^+$ axons (PV$^+$SMI 312$^+$) and total axons (SMI 312$^+$) (normalized to ctl) (ctl = 6 mice, cKO = 4 mice, two-sided unpaired $t$-tests). Immunostaining (**f**) and quantification (**g**) of PV$^+$ interneurons in different layers of mPFC at the age of 9 weeks (ctl = 6 mice, cKO = 9 mice; two-way Anova, Sidak's multiple comparison for different layers; two-sided unpaired $t$-tests for total). **h** Ratio of PV$^+$ interneuron axons to all axons. (ctl = 6 mice, cKO = 4 mice, two-sided unpaired $t$-test). Data are shown as mean ± SEM in **e**, **g** and **h**. Source data are provided as a Source Data file.

potential morphological changes of PV axons, e.g., thinner axonal caliber and/or shortened axons in the cKO mPFC.

Together, our results demonstrated that OPC-GABA$_B$R were essential for the correct cell density and myelination of interneurons in the mPFC.

**Surplus of interneurons display suppressed activity in the cKO mPFC.** To investigate the impact of the supernumerous interneurons detected in the mutant mPFC for the local network activity, we recorded spontaneous inhibitory postsynaptic currents (sIPSCs) and sEPSCs of the pyramidal neurons in layer V of the mPFC (Fig. 3a, b), where interneurons are abundantly located (Fig. 2f, g). In contrast to our expectation, the increase of PV interneuron density did not result in a concomitant increase of inhibitory input. In contrast, the frequency of sIPSCs was reduced by 40% (Fig. 3c) while simultaneously the amplitude of sIPSCs

remained unaffected in the mutant mPFC (Fig. 3d), as well as the firing rate and amplitude of sEPSCs (Fig. 3e, f).

Neuronal activity can regulate myelination[23,27]. Therefore, considering the attenuated interneuron myelination and decreased inhibitory tone in the cKO mPFC, we asked whether such morphological and physiological changes could already be initiated during early development and before myelination onset at p14 in mPFC (Supplementary Fig. 7). To answer this question, we assessed the density of the vesicular GABA transporter (vGAT) (Fig. 3g, h) as a readout of inhibitory neuron activity[28]. Indeed, already at p10, i.e., only 3 days after the first tamoxifen injection, the density and the volume of vGAT immunopuncta were reduced to 75% in the cKO mPFC (55.4 to 41.8 puncta/ $1 \times 10^{-3}$ mm$^3$ and 1.42 to 1.05 µm$^3$, respectively) (Fig. 3h–j, Supplementary Fig. 8a–d), suggesting a reduced interneuron activity in the cKO mPFC. Since the expression of immediate early genes has been correlated with the electrical activity of

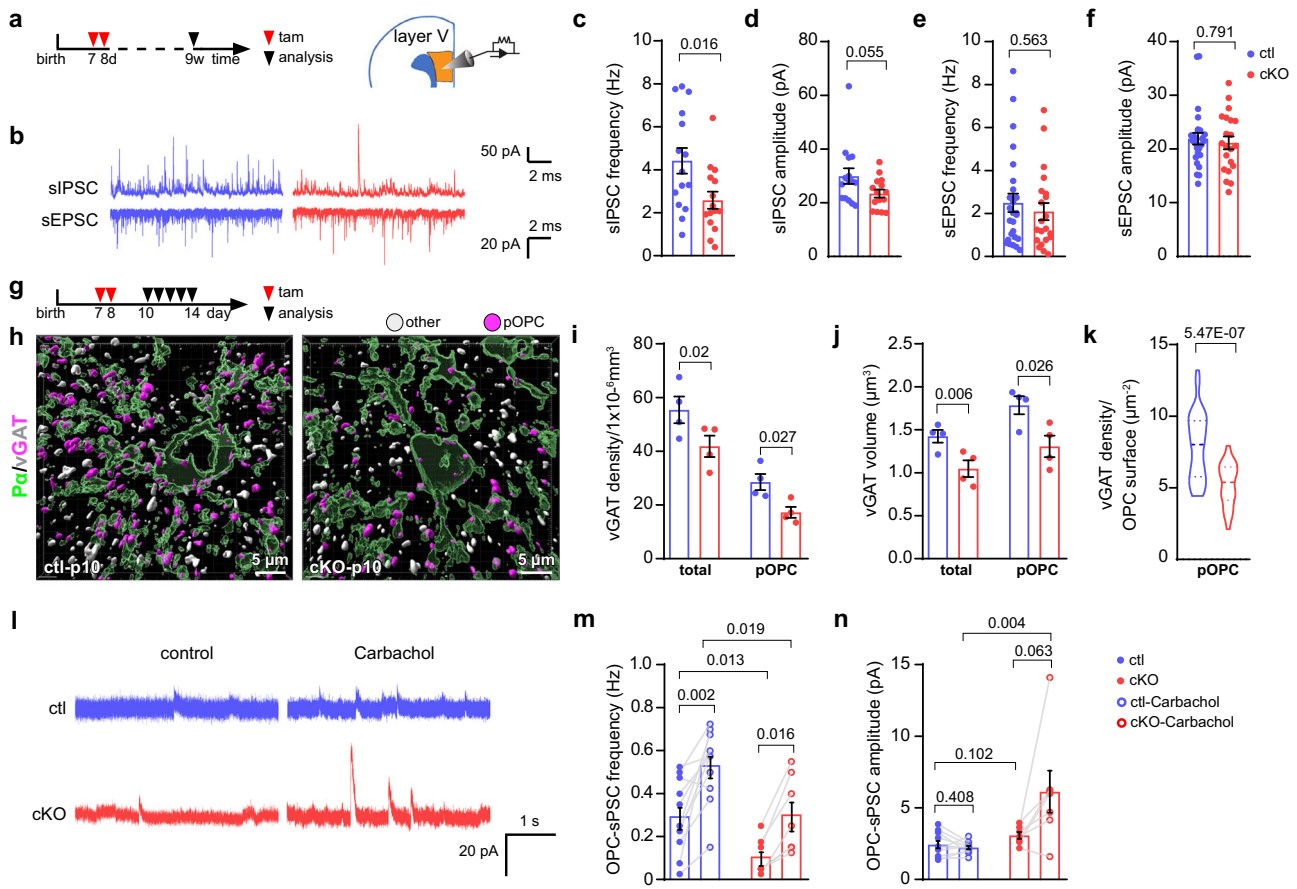

**Fig. 3 Interneuron activity is impaired in the mPFC of OPC-GABA_BR cKO mice. a** Experimental schedule for **b–f** and the scheme of analyzed brain region (orange). **b** Patch-clamp recordings of sIPSCs and sEPSCs in pyramidal neurons of layer V of medial prefrontal cortex (mPFC). **c–f** Quantification of frequencies and amplitudes of spontaneous inhibitory and excitatory postsynaptic currents (sIPSC and sEPSC, respectively) (sIPSC: ctl = 15 cells from 4 mice, cKO = 15 cells from 5 mice; sEPSC: ctl = 26 cells from 4 mice, cKO = 21 cells from 5 mice; sIPSC frequency was analyzed with two-sided unpaired *t*-test, **d–f**: two-sided unpaired Mann–Whitney test). **g** Experimental schedule for **h–n**. **h** Surface renderings of OPCs and inhibitory presynapses based on PDGFRα (Pα, green) and vGAT (vesicular GABA transporter, white and magenta) immunostaining in ctl and cKO mPFC at p10 using Imaris software. vGAT was classified into two subgroups: on putative OPC synapses (pOPC, magenta vGATs with less than 200 nm distance from the OPC surface) and other synapses (other, gray vGATs more than 200 nm). Density (**i**) and volume (**j**) of vGAT in ctl and cKO mPFC. (ctl = 4 mice, cKO = 4 mice; two-sided unpaired *t*-tests). **k** Quantification of synaptic vGAT (<200 nm) density per μm$^2$ of OPC surface. (ctl = 4 mice, cKO = 4 mice, two-sided unpaired *t*-tests). **l** Patch-clamp recordings of spontaneous postsynaptic currents (sPSCs) of OPCs at the layer of V of mPFC at p11-14. **m, n** Quantification of sPSC frequency and amplitude of OPCs before and after addition of carbachol. (ctl = 11 cells from 3 mice, cKO = 7 cells from 3 mice, two-sided unpaired *t*-tests). Data are shown as mean ± SEM in **c–f**, **i**, **j**, **m** and **n**. In **k**, data are shown with indications of median and quartiles in thick and thin dashed lines, respectively. Source data are provided as a Source Data file.

neurons[29,30], we investigated the expression of cFos in PV interneurons at p10 and p14. Although at p10, we observed already a reduction of vGAT in the cKO mPFC (Fig. 3i, j), the proportions of PV$^+$cFos$^+$ cells were comparable in the cKO and ctl mPFC (Supplementary Fig. 8f). However, at p14, the density and the percentage of PV$^+$ interneurons with cFos expression was reduced by 60–75% in the cKO mPFC (0.35 to 0.14 PV$^+$cFos$^+$ cells/1 × 10$^{-3}$ mm$^3$; 4.4% to 1.1% PV$^+$cFos$^+$/PV$^+$ cells; Supplementary Fig. 8g). These results suggested that the activity of inhibitory neurons was impaired in the cKO mPFC during development.

To demonstrate whether such a change of interneuron activity could affect GABAergic communication between interneurons and OPCs, we further analyzed the vGAT density on OPCs. For that purpose, we classified the puncta of the vGAT immunolabel into two subgroups based on the distance between the vGAT label and the OPC surface using the image analysis software Imaris (Fig. 3h). We defined the vGAT puncta with a 'distance < 200 nm' as vesicles to be potentially released to OPCs (magenta, pOPCs),

and vGAT puncta with a 'distance > 200 nm' as vesicles targeted to other cells (gray, other cells), based on the assumption of about 270 nm as the distance of presynaptic vGAT and postsynaptic GABA_AR/gephyrin[31]. The density and the volume of vGAT immunolabel close to OPCs were decreased (Fig. 3i, j), including less puncta at the OPC surface in cKO mPFC (Fig. 3k), suggesting a reduction of GABAergic input to OPCs in the cKO mPFC. To confirm this observation, we took advantage of the GABA_A receptor expression (in addition to GABA_BR) of OPCs and evaluated the frequency of spontaneous GABAergic currents in OPCs (OPC-sPSC) of p11–14 mice (Fig. 3l). Our results showed that the OPC-sPSC frequency was reduced by about 64% (0.3 to 0.1 Hz) in the cKO mPFC (Fig. 3m), while the amplitude remained unaffected (Fig. 3n), indicating OPCs receive less GABAergic signals in the mutant mouse brain. In this context, it is important to remind that GABAergic currents of OPCs are depolarizing (due to high intracellular Cl$^-$) with the potential to activate voltage-gated Ca$^{2+}$ channels in OPCs[32,33]. In addition, we increased the neuronal firing by muscarinergic stimulation

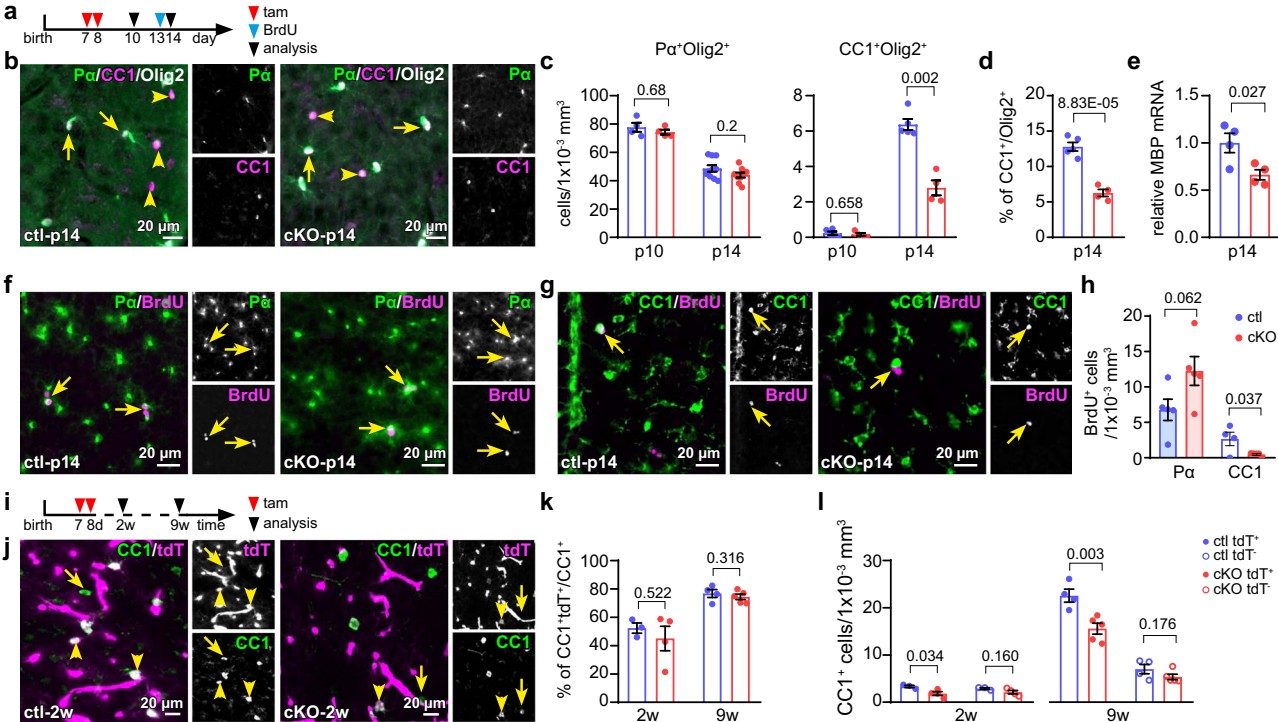

**Fig. 4 Reduced oligodendrocyte differentiation and myelin gene expression at the onset of myelination in the mutant mPFC. a** Experimental plan. **b** Immunostaining of OPCs and oligodendrocytes (OLs) for PDGFRα (Pα, green), Olig2 (white) and CC1 (magenta) in p14 medial prefrontal cortex (mPFC). **c** Density of OPCs and OLs in ctl and cKO mPFC at p10 and p14. (OPC: p10: ctl = 4 mice, cKO = 4 mice; p14: ctl = 10 mice, cKO = 9 mice; OL: p10: ctl = 5 mice, cKO = 4 mice; p14: ctl = 5 mice, cKO = 4 mice, two-sided unpaired t-tests). **d** Percentage of OLs of the complete lineage at p14 (ctl = 5 mice, cKO = 4 mice, two-sided unpaired t-tests). **e** Quantitative analysis of MBP mRNA level in ctl and cKO mPFC at p14. (ctl = 4 mice, cKO = 4 mice, two-sided unpaired t-test). Immunostaining (**f**, **g**) and quantification (**h**) of OPCs (Pα+, green in **f**) and OLs (CC1+, green in **g**) incorporated with BrdU (magenta). (OPC: ctl = 5 mice, cKO = 5 mice, two-sided unpaired t-test; OL: ctl = 4 mice, cKO = 5 mice, two-sided unpaired t-test). **i** Experimental design of **j–l**. **j** Immunostaining of OLs (CC1+, green) combined with tdTomato (tdT, magenta) expression in the mPFC at 2w. **k** Recombination efficiency of OLs. (2w: ctl = 3 mice, cKO = 4 mice, two-sided unpaired t-test; 9w: ctl = 4 mice, cKO = 5 mice, two-sided unpaired t-test). **l** Comparison of recombined and non-recombined OL densities (with or without tdT expression). (2w: ctl = 3 mice, cKO = 4 mice, two-sided unpaired t-tests; 9w: ctl = 4 mice, cKO = 5 mice, two-sided unpaired t-tests). Data are shown as mean ± SEM in **c–e**, **h**, **k** and **l**. Source data are provided as a Source Data file.

using carbachol[34] and observed a rise in OPC-sPSC frequency (Fig. 3l, m). Also under these conditions, we still recorded a 40% lower frequency in the cKO OPCs. In addition, with carbachol treatment, we observed more variability of slightly higher current amplitudes in cKO OPC (Fig. 3n), which we attributed to putative changes of vesicle loading upon presynaptic muscarinic stimulation. Since the current amplitudes evoked by application of GABA were unperturbed in the cKO OPCs (Supplementary Fig. 8e), the expression of GABA$_A$ receptors appeared unaffected by the GABA$_B$R ablation. Therefore, our data strongly suggested an impairment of interneuron activity in the mutant mouse mPFC.

In summary, we concluded that interneurons exhibit lower activity and transmit less GABAergic signals to OPCs in the cKO mPFC during development.

**Decreased OPC differentiation in the cKO mPFC during development**. Next, we asked whether the reduced inhibitory tone could contribute to the observed changes in OPC differentiation and whether it could correlate with the onset of myelination specifically. For that purpose, we compared the densities of OPCs and OLs populations at p10 and p14 (Fig. 4a–c). At p10, the densities of both were still comparable in the cKO and ctl mPFC. However, at p14, the differentiation and cell numbers of OLs dropped strongly in the mutant mPFC (Fig. 4c, d), as did the MBP expression (Fig. 4e). As the PV/cFos immunostaining

results had shown, the interneuron activities of the cKO mPFC was still comparable to the ctl mPFC at p10 while it was suppressed at p14 (Supplementary Fig. 8f, g), thereby suggesting that interneuron activity and OPC differentiation are co-regulated during development. To confirm that the reduction of the OL density was attributed to the OPC differentiation but not to OL apoptosis, we performed a BrdU proliferation/differentiation assay and immunostaining for cleaved caspase-3 (CC-3, a well established marker for apoptosis) at p14. BrdU was administered at p13 to label proliferating cells, and we analyzed the population of cells incorporated with BrdU at p14 (Fig. 4a). The density of Pα+BrdU+ cells did not change at p14, however, the number of CC1+BrdU+ cells was drastically reduced by 85% (2.7 to 0.5 cells/ 1 × 10$^{-3}$ mm$^3$, Fig. 4f–h). In contrast, the CC-3 staining showed a similarly low apoptosis rate of OLs in both groups at p14 (Supplementary Fig. 9). Again, in the corpus callosum, the populations of OPCs and OLs were not affected by the deletion of GABA$_B$Rs at p14 (Supplementary Fig. 3). These results suggested that reduced GABAergic input gave rise to attenuated OPC differentiation in the cKO mPFC during development.

When we induced GABA$_B$R deletion in the young adult mice at the age of 4 weeks, representing the peak of OPC-GABA$_B$R expression during development (Supplementary Fig. 1c), the density of OLs as well as the MBP expression were not affected in the cKO mice both in the mPFC and corpus callosum (Supplementary Fig. 10). These data strongly suggested that the

OPC-GABA$_B$R was not directly involved in OPC differentiation, at least during early postnatal weeks. In addition, the percentage of recombined tdT$^+$ OLs of all OLs in both cKO and ctl mPFC did not differ at 2w and 9w (Fig. 4i–k), despite decreased densities of mature CC1$^+$ OLs (Figs. 4c and 1g) or CC1$^+$tdT$^+$ OLs (Fig. 4l) in the cKO mPFC. Thereby, it implied that the reduced OPC differentiation during development was rather due to the impaired interneuron-OPC communication.

In summary, our data demonstrated that during early development of the mPFC, deletion of GABA$_B$R in OPCs affected the inhibitory tone of interneurons, feeding back onto OPC differentiation and interneuron myelination.

**OPCs induce interneuron apoptosis via GABA$_B$R-TWEAK signaling**. As we had seen a surplus of interneurons in the mutant mPFC which could explain the abnormal neuronal activity[8,35], we tested whether the interneuron numbers would be accurately controlled by programmed cell death early during development. After induction of GABA$_B$R in OPCs at p7 and 8, the extent of interneuron apoptosis was determined at p10 and p14 by immunostaining for PV and CC-3 (Supplementary Fig. 11). As expected by the earlier experiments, we found a 50% reduction of PV$^+$/CC-3$^+$ interneurons at p10 suggestive of a strongly reduced apoptosis of PV$^+$ interneurons in the mutant mPFC (Supplementary Fig. 11). These results indicated a defective apoptotic process of the interneurons in the mPFC of OPC-GABA$_B$R cKO mice during development.

To confirm the selective cell death of interneurons, we already induced the OPC-GABA$_B$R deletion as early as p1 and p2 and compared the apoptosis of inhibitory neurons with that of excitatory ones at p5 and p7 (Fig. 5a). The interneuron apoptosis lasted from p1 till p15 with a peak at p7[7], while the time window for the excitatory neuron apoptosis was rather narrow, between p2–5[36]. The apoptosis of total interneurons (GAD67$^+$CC-3$^+$) as well as of PV interneurons (PV$^+$CC-3$^+$, Fig. 5b) were significantly reduced in cKO mPFC at p5 and p7 (Fig. 5c), while that of the excitatory neurons (CTIP$^+$CC-3$^+$, Fig. 5b; TBR1$^+$CC-3$^+$, Fig. 5c) did not change at both time points (Fig. 5c, Supplementary Fig. 12). Therefore, we concluded that OPCs sent pro-apoptotic signals back to interneurons at the early postnatal weeks.

To molecularly identify the critical apoptotic factor released by OPCs, we tested the expression levels of six peptides which had previously been associated with cell survival or cell death. Gene deletion was induced at p1 and 2, and mPFC was collected at p7, the peak of interneuron apoptosis (Supplementary Fig. 13a). Out of the six cytokines, only the expression of TWEAK (tumor necrosis factor-like weak inducer of apoptosis, also known as Apo3l or TNF superfamily member 12, TNFSF12) was significantly changed with a reduction of about 30% in cKO mPFC (Fig. 5d, Supplementary Fig. 13b). Subsequently, we tested MACs-purified OPCs (collected from cKO and ctl cortices at p7) (Fig. 5d) as well as Oli-neu cells (an OPC cell line) treated with GABA$_B$R antagonist CGP 55845 for TWEAK expression. Also here, we observed a downregulation of TWEAK (Fig. 5d). In three independent experiments, including genetic deletion or pharmacological inhibition of OPC GABA$_B$R signaling, we consistently observed reduced levels of TWEAK (Fig. 5d, Supplementary Fig. 13).

To confirm that OPC-derived TWEAK can indeed induce neuronal apoptosis after binding to its cognate TWEAK receptor (TWEAKR; TNF Receptor Superfamily Member 12A, TNFRSF12A; also known as Fibroblast Growth Factor-inducible 14 or FN14)[37], we first verified the expression of TWEAKR on the PV neurons in the mPFC at p5 by immunohistochemistry

(Fig. 5e). Then, we treated primary cortical neurons with conditioned medium obtained from OPC Oli-neu cultures (Fig. 5f). The selective TWEAKR antagonist, a triazolyl-thiomorpholinyl-methanone L524-0366, was added to the conditioned medium at 20 μM competing for the binding of TWEAK to the TWEAKR on interneurons (Fig. 5f, g). While conditioned medium harvested from the OPC cell line increased the interneuron apoptosis, blocking TWEAKR signaling by co-incubation with the antagonist L524-0366 prevented cell death (Fig. 5h, i). These results further substantiated that OPCs could induce interneuron apoptosis via TWEAK secretion.

In summary, our results demonstrated that OPCs induced interneuron apoptosis by releasing TWEAK upon GABA$_B$R activation. Blocking this pathway prevented the apoptosis of interneurons. In the mPFC, the concomitant dysregulation of interneuron density resulted in reduced inhibitory GABAergic tone and altered myelination.

**Interrupted OPC-interneuron communication generates cognitive impairment**. The mPFC is responsible for cognitive processes and their regulatory finetuning is guaranteed by the E/I balance[38]. To test whether the selectively reduced interneuron apoptosis and the associated structural myelin alterations could generate defects in neural circuits and cognition, we performed in vivo electroencephalographical (EEG) recordings and behavioral tests in 9w old mice after gene ablation (induced at p7 and 8, Fig. 6a).

Our patch-clamp recordings had already revealed a reduced interneuron activity suggesting an imbalance of E/I in the mutant mPFC (Fig. 3b–f). And indeed, an impaired cortical network activity of the mutant mice was also detected in EEG recordings. The relative contribution of the gamma band power (30–80 Hz) to the total brain oscillatory pattern, in particular, the lower gamma wave band (30–50 Hz) was slightly, but significantly decreased by about 10% in cKO mice (Fig. 6b, c, Supplementary Fig. 14a, b).

To evaluate the impact of the circuit dysregulation in the mutant mPFC for the living animal, we challenged mice for their cognitive performance employing tasks of social cognitive behavior which had been associated with mPFC function, i.e., social novelty and new object recognition[39]. In addition, nest building behavior is often introduced as an indicator of general well-being, and its alteration as an early sign of cognitive decline[40]. We observed that the mutant mice built significantly less elaborated nests, even not reaching half the performance score of ctl mice, indicating putative cognitive deficits (Fig. 6d). The three-chamber test was used to assess social cognition by analyzing mice for a form of general sociability and their interest in social novelty (Fig. 6e). First, the three-chamber test was used to monitor general social behavior, i.e., preference of other mice over objects. Both, ctl and cKO mice exhibited more interest (i.e., longer sniffing times) into the mouse rather than the object during the first test phase, indicating no disorder in sociability of the mutant mice (Fig. 6f, Supplementary Fig. 14c, d, g, h). At the second phase, a social novelty preference was assessed. While control animals explored the unfamiliar mouse longer than the familiar one, the cKO mice could not discriminate stranger and familiar mouse (Fig. 6g, Supplementary Fig. 14e, f, i, j). Also, when we employed the novel object recognition test (Fig. 6h), the cKO mice did not recognize a new object when it was replaced after a 40 min break from the habituation (Fig. 6i, j). The general motor activity appeared only mildly affected. In the open-field the motor activity of the cKO mice was slightly increased based on the total distance run in the cage (Fig. 6k–m).

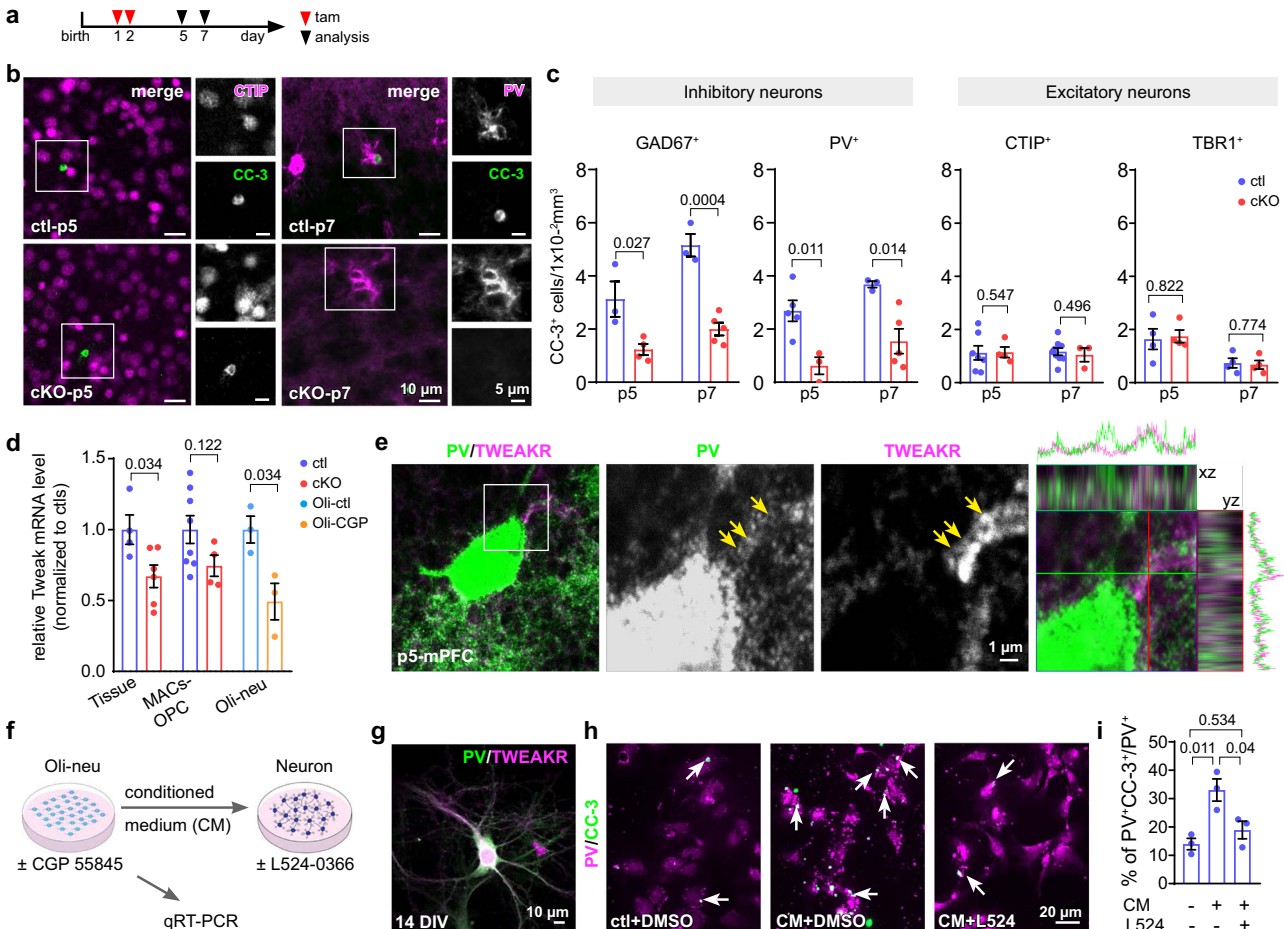

**Fig. 5 During early CNS development impaired release of TWEAK from GABA_BR-deficient OPCs mitigated programmed cell death of interneurons.**
**a** Experimental schedule. **b** Fluorescent micrographs of excitatory (CtBP-interacting protein (CTIP)+, magenta) and inhibitory (parvalbumin (pv)+, magenta) interneurons co-stained with the apoptotic marker cleaved caspase 3 (CC-3, green) in p5 and p7 mPFC. **c** Quantification of apoptotic neurons (CC-3+) co-expressing the interneuron marker glutamate decarboxylase 67 (GAD67) and PV or CTIP and T-box brain transcription factor 1 (TBR1) as markers for excitatory neurons. (GAD67: p5-ctl = 3 mice, p5-cKO = 4 mice; p7-ctl = 3 mice, p7-cKO = 5 mice) (PV: p5-ctl = 5 mice, p5-cKO = 3 mice; p7-ctl = 3 mice, p7-cKO = 5 mice) (CTIP: p5-ctl = 7 mice, p5-cKO = 4 mice; p7-ctl = 9 mice, p7-cKO = 3 mice) (TBR1: p5-ctl = 4, p5-cKO = 4; p7-ctl = 4, p7-cKO = 4 mice) (two-sided unpaired t-tests). **d** Relative mRNA level of TNF like weak inducer of apoptosis (TWEAK) in the medial prefrontal cortex (mPFC), magnetic-activated cell sorted (MACs)-OPCs from ctl and cKO mice as well as Oli-neu cells treated with or without 20 μM CGP 55845 (tissue-ctl = 4 mice, tissue-cKO = 6 mice; MACs-ctl = 8 mice, MACs-cKO = 4 mice; Oli-neu: n = 3 independent experiments; two-sided unpaired t-tests). **e** Double-immunostaining of TWEAK receptor (TWEAKR, magenta) and PV (green) in the control mPFC at p5. Arrows indicate co-expression of TWEAKR and PV. **f** Experimental design for in vitro studies. **g** Immunolabeling of TWEAKR (magenta) on 14 days in vitro (DIV) PV+ interneurons (green). **h** Immunostaining of apoptotic primary PV+ neurons with CC-3 (green) and PV (magenta) after being treated with conditional medium (CM) of Oli-neu cells co-treated with or without TWEAKR antagonist L524-0366 (L524, 20 μM). White arrows indicate PV+ cells co-expressing CC-3. **i** Percentage of apoptotic PV+ interneurons of all PV+ interneurons in conditioned medium and treated with a competitive TWEAKR inhibitor (n = 3 independent experiments from distinct samples, one-way ANOVA, Turkey's multiple comparison). Data are shown as mean ± SEM in **c**, **d**, and **i**. Source data are provided as a Source Data file.

Hence, we conclude that GABA_BRs of OPCs are essential for the fine-tuning of inhibitory circuits in the mPFC, and thereby for distinct cognitive behavior.

## Discussion

A balance of excitation and inhibition (E/I) in the medial prefrontal cortex (mPFC) is of key importance for mammalian cognition. Establishing correct cell densities and subsequent myelination have been identified as pivotal elements during CNS development[35,41]. This has been well documented for fast-spiking parvalbumin-positive (PV+) interneurons[6,17]. Here, we highlight OPCs as indispensable regulators of the interneuron population and its myelination in the mPFC. We provide a molecular explanation and can demonstrate how impaired

bidirectional OPC-interneuron signaling affects social cognitive behavior. At the first two postnatal weeks OPCs sense the interneuron activity via expression of GABA_B receptors and concomitantly adjust interneuron cell density by releasing TNF-like weak inducer of apoptosis (TWEAK) limiting interneuron survival. When the interneuron-OPC signaling is interrupted by genetic ablation of GABA_BR in OPCs and attenuated release of TWEAK, the population of fast-spiking, parvalbumin-positive interneurons builds up. However, despite their increased cell numbers, the interneurons appear hypoactive, have less contacts with OPCs during development and exhibit deteriorated myelin structures in the adulthood. Notably, the mutant mice exhibit severe cognitive defects in their social behavior.

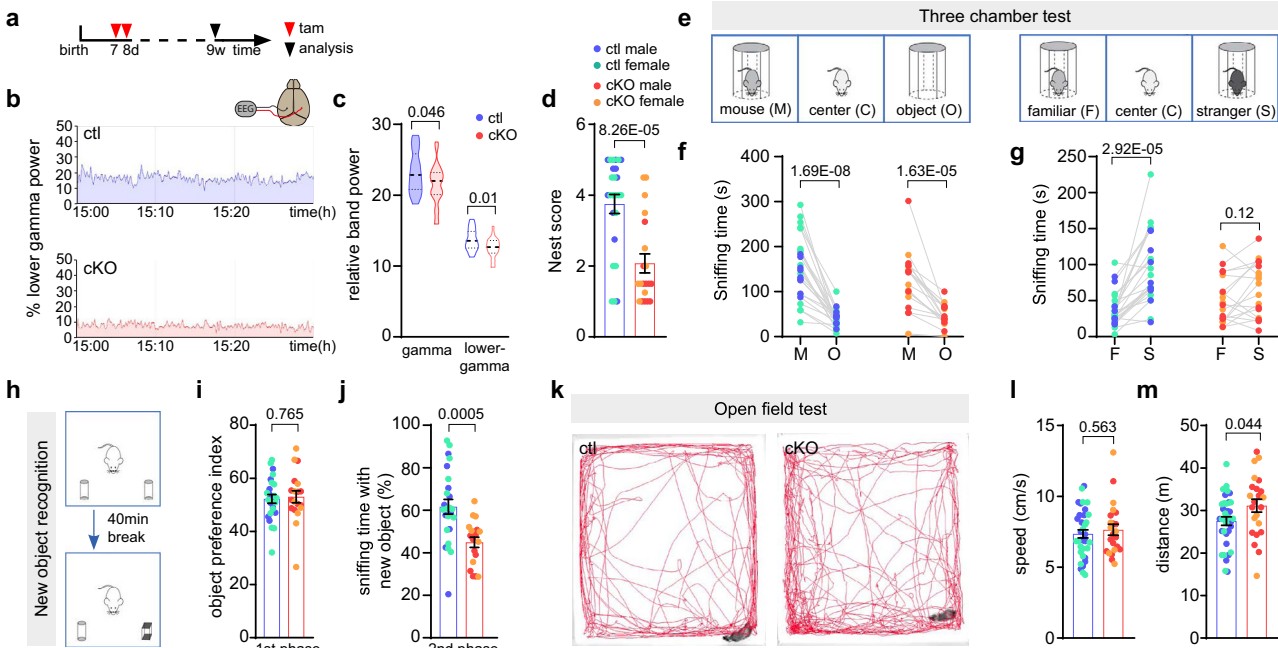

**Fig. 6 In OPC-GABA$_B$R cKO mice a perturbed neuronal firing translates into deficits of social behavior. a** Experimental schedule. **b** Lower gamma band acquired by electroencephalogram recordings. **c** Violin plot shows decreased gamma oscillation in the cKO mouse brain. (ctl = 24 h from 4 mice, cKO = 24 h from 4 mice, two-sided unpaired $t$-tests). **d** Impaired nest building ability of mutant mice. (male: ctl = 10 mice, cKO = 10 mice; female: ctl = 15 mice, cKO = 10 mice, two-sided unpaired $t$-test). **e** Scheme of the three-chamber behavior test. **f** Quantification of the sniffing time with mouse (M) and object (O) (male: ctl = 9 mice, cKO = 9 mice, two-sided paired $t$-tests; female: ctl = 15 mice, cKO = 9 mice, two-sided paired $t$-tests). **g** Quantification of the sniffing time with familiar (F) and stranger (S) mouse (male: ctl = 9 mice, cKO = 9 mice, two-sided paired Wilcoxon test; female: ctl = 15 mice, cKO = 9 mice, two-sided paired $t$-test). **h** Scheme of new object recognition test. **i** Assessment of object preference (male: ctl = 10 mice, cKO = 10 mice; female: ctl = 15 mice, cKO = 9 mice, two-sided unpaired $t$-test). **j** Percentage of sniffing time with new object among total sniffing time. (male: ctl = 10 mice, cKO = 10 mice; female: ctl = 15 mice, cKO = 9 mice, two-sided unpaired $t$-test). **k** Representative trajectory charts of the animals during the open field test. **l, m** Both ctl and cKO mice exhibited similar motor activities shown by the speed and distance analysis. (male: ctl = 17 mice, cKO = 14 mice; female: ctl = 18 mice, cKO = 9 mice, two-sided unpaired $t$-test). Data are shown as mean ± SEM in **d, i, j, l** and **m**. In **c**, data are shown with indications of median and quartiles in thick and thin dashed lines, respectively. Source data are provided as a Source Data file.

During development, large populations of immature interneurons invade the cortex where their cell density is adjusted by programmed cell death[9,10]. For assuring their survival, these neurons form connections with adjacent cells and receive retrograde signals from their targets[10,11]. Interneuron apoptosis drastically increases at p7[7], shortly after formation of interneuron-OPC synapses at p4–5[12], and it stays elevated till p11[7], right after the peak of interneuron-OPC connectivity at p10[13]. Based on this established knowledge, we hypothesized that OPCs could exert an apoptotic impact to their presynaptic partners. Indeed, our data provide strong evidence that, starting from postnatal day 5, OPCs elicit an apoptotic cascade of adjacent interneurons to adjust their density. In OPCs, the regulatory process employs a GABA$_B$R-TWEAK signaling pathway. TWEAK, TNF super family member 12 (TNFSF12), is TNF-like weak apoptotic factor. Membrane-bound and cleaved isoforms of TWEAK are both able to induce cell apoptosis by binding to the TWEAK receptor (TWEAKR, also known as fibroblast growth factor-inducible 14 or FN14) on the neuronal side[37,42,43]. According to our data GABA$_B$R-TWEAK signaling preferentially affects interneurons, suggesting this 'kill me' signal acting highly localized. Indeed, TWEAK is specifically recruited to synapses where TWEAKRs are expressed at the neuronal membrane[44]. OPCs directly sense GABAergic signals at their interneuron-OPC synapse. We suggest that subsequent signaling triggers TWEAK synthesis and translocation to the OPC surface, where it can directly bind interneuronal TWEAKRs. Alternatively, the TWEAK ectodomain could be proteolytically cleaved and act as a soluble factor. However, this scenario could also happen at soma-somatic contact sites between OPCs and interneurons[2,45,46]. Overall, the regulatory impact appears to be highly specific since the OPC-induced interneuron apoptosis was restricted to the first two postnatal weeks, a well-established time window of interneuron apoptosis[7,36]. This also explains why the cKO induction at the age of 4w did not affect OPC differentiation and myelination, although GABA$_B$R expression peaks in OPCs at 4w (Supplementary Fig. 1c). Please note, instead of a gradual decline, we found a slight increase of PV$^+$CC3$^+$ cells at p14 compared to p10 in both control and mutant mice, probably due to the age-dependent increase of PV expression described for fast spiking interneurons[47].

GABA$_B$Rs are G-protein coupled receptors. In cultures of OPCs, activation of GABA$_B$R negatively regulates adenylyl cyclase and reduces cAMP levels[48]. Subsequently, protein kinase A activity is suppressed, followed by impeded nuclear translocation of cAMP response element binding protein (CREB) affecting gene expression, e.g., the expression of brain derived neurotrophic factor (BDNF) or AMPA (α-amino-3-hydroxy-5-methyl-4-isoxazolepropionic acid) -type glutamate receptor GluA1 subunit[49,50]. A recent study suggested that activation of GABA$_B$R in cultured OPCs can also activate Akt/Src kinases required for OPC differentiation[14]. Additional studies are necessary to elucidate which class of G proteins (G$_{αi}$ and/or G$_{βγ}$) transmit GABA$_B$R signaling in OPCs to elicit TWEAK expression and release. As shown by RNA sequencing, microglia express rather high levels of TWEAKR during development in the cortex[51]. However, whether

or not a TWEAK-based OPC-microglial crosstalk contributes to the elimination of interneurons during development is not clear yet. At this point, we can not exclude a contribution of TWEAKR-expressing microglia, which requires future experiments. So far, our in vitro data show that in a pure culture system, TWEAKR antagonist can interfere with interneuron apoptosis induced by TWEAK from conditioned medium of OPCs (Oli-neu cell line).

A neuronal surplus is often accompanied with reduced activity[6,8]. In the OPC-GABA$_B$R cKO mice, we observed a strong increase of PV$^+$ interneurons to about 125% of ctl levels. However, their presynaptic terminals contained less immunolabel for the vesicular GABA transporter (vGAT) and implying a significant reduction in activity and GABA release. The reduction of vGAT puncta to 75% represents a reduction of about 60% for PV interneurons (i.e., 75% of 125%), as observed for the PV$^+$cFos$^+$ cell density (63%) or the sIPSC frequency (64%). It is quite conceivable that the surviving, supernumerous PV interneurons are physiologically less mature in the cKO mPFC[52]. Additionally, a potential change in the PV interneuron morphology is detected since the relative axon volumes of PV interneurons in cKO and ctl mice were identical despite a larger population of PV$^+$ interneurons in the mPFC of mutant mice, e.g., thinner axonal caliber, shortened axons or less arborized axons in the cKO mPFC. However, such a detailed anatomical analysis is beyond the scope of this study.

Interestingly, also the early loss of the ionotropic GABA$_A$R γ2 subunit in OPCs reduced the firing rate of presynaptic PV$^+$ interneurons as well as a similar myelin defect[4]. The very different signal pathways of ionotropic and metabotropic GABA receptors merge and similarly affect PV interneuron activity, OPC-axon contacts and myelin gene expression. However, the impact on myelin structures was rather distinct. GABA$_A$R deletion resulted in prolonged lengths of nodes and internodes with a decreased density of paranodes[4], while only an extended length of paranodes was detected in the OPC-GABA$_B$R cKO mPFC of our study. In addition, GABA$_A$Rs facilitated interneuron maturation in the juvenile (p24) somatosensory cortex, while GABA$_B$Rs regulate the elimination of interneurons during the first two postnatal weeks. Apparently, both receptors optimize the density and activity of PV interneurons, cooperating jointly but employing different machineries at distinct time points.

The GABA$_A$R signaling between interneurons and OPCs is relatively well established[4,15,53,54]. E.g., the γ2 subunit of GABA$_A$R contributes to the maintenance of the OPC, but has no impact on OL formation[15]. However, the role of GABA$_B$Rs for OPC differentiation is complex. We observed that GABA$_B$R was essential for formation of OLs at the early development but not at later stages. Similarly, an in vitro study showed that activation of GABA$_B$R by the selective agonist baclofen could stimulate the differentiation of cortical OPCs prepared from p0–2 pups[14]. Since OPCs are heterogeneous with age[55], GABA$_B$R of OPCs could contribute to the developmental diversity. Differentiation of OLs and subsequent myelination are strongly affected by neuronal activity and axonal caliber[23,27,56]. Indeed, we observed a suppression of the inhibitory tone in the mPFC at p10 followed by a change in oligodendrocyte density detected at p14. Concomitantly, the oligodendrocytes formed less myelin (only 40% of control) around the interneuronal axons and altered paranodal structures in the adult mPFC, which are essential for precise action potential propagation. The elongation of paranode will be accompanied with redistribution of Kv1 channels from under juxtaparanode towards paranode in the brain of multiple sclerosis and aging[57]. These exposed Kv1 channels are more active. As a result, these changes hinder the action potential propagation, which in turn reduces the synaptic communication. Regardless, the morphological deterioration was only observed in the mPFC,

but not in the corpus callosum, largely composed axons of excitatory neurons only[22]. In line with these neuron-type specific observations, the qRT-PCR results indicated reduced levels of MBP, but neither PLP nor MOG mRNA expression in the mutant mPFC, both at p14 and 9w of age. Myelin proteins of inhibitory and excitatory axons are differently expressed, e.g., MBP is preferentially expressed in the myelin of inhibitory axons[16].

The GABAergic signaling of OPCs plays pivotal roles for the myelination of interneurons, which ensures precise finetuning of the local neural circuitry[58]. As seen in chandelier cells[6], excessive number of interneurons in the OPC-GABA$_B$R cKO mice exhibited suppressed activity and reduced myelination in the mPFC. All these physiological and morphological abnormalities certainly contribute to the cognitive impairment. In early socially isolated mice the observed cognitive impairment is also accompanied by a strong hypomyelination in the mPFC[59,60]. A proper E/I ratio in the prefrontal cortex, especially the E/I balance in the postnatally developing mPFC, is extremely important for social cognition[38,61]. When the pyramidal neuron activity was enhanced in the mPFC at early p7–11 caused a severe cognitive dysfunction in the adulthood[61]. In addition, despite relatively short axons, myelination of cortical interneurons (mainly PV$^+$ neurons[16]) contributes significantly to the finetuning of local activity in the mPFC and cortex, insuring proper behavior performance[4,6,17]. Nota bene, since the genetic manipulation is not selective to the PFC, we cannot rule out contributions of other brain regions to the behavioral phenotype, which has to be left for future studies.

In conclusion, our study demonstrates that OPCs regulate inhibition in the mPFC via GABA$_B$R/TWEAK signaling. During development, OPCs determine the density of interneurons by adjusting their apoptosis. Subsequently, correct myelination of interneurons, as an essential component of the network activity in the mPFC, determines the cognition and social behavior.

## Methods

**Ethics statement.** Animal husbandry and procedures were performed at the animal facility of CIPMM, University of Saarland according to European and German guidelines for the welfare of experimental animals. Animal experiments were approved by the Saarland state's "Landesamt für Gesundheit und Verbraucherschutz" in Saarbrücken/Germany (animal license number: 65/2013, 12/2014, 34/2016, 36/2016, 03/2021 and 08/2021).

**Animals.** To conditionally knock out (cKO) GABA$_B$ receptor subunit1 in oligodendrocyte precursor cells (OPCs), TgH(NG2-Cre$^{ERT2}$)[19] mice were crossbred to GABA$_{B1}$$^{lox511/lox511}$ mice (flanking exon 7 and 8 of *gabbr1*)[62]. Mice with genotypes of NG2$^{ct2/wt}$ x GABA$_{B1}$R$^{fl/fl}$ were used as cKO and the littermates NG2$^{wt/wt}$ x GABA$_{B1}$R$^{fl/fl}$ or NG2$^{ct2/wt}$ x GABA$_{B1}$R$^{wt/wt}$ were controls after tamoxifen application. To visualize the recombined cells, we crossbred the double transgenic mice with TgH(Rosa26-CAG-$^{fl}$STOP$^{fl}$-tdTomato) (Rosa26-tdTomato)[63]. Mouse strains were maintained in C57Bl/6 N background. Mice were kept at the animal facility of the CIPMM in a 12 h light/dark cycle at 20 °C with humidity at 55–70% and fed a breeding diet (V1125, Sniff) *ad libitum*.

Mouse numbers and ages are indicated in the main text and figure legends. Behavioral tests were carried out at the age of 9 weeks. Both genders were used in all experiments.

**Tamoxifen administration.** Tamoxifen was dissolved in Miglyol (3274, Caesar&Loretz GmbH, Hilden, Germany) to a final concentration of 10 mg/ml. Tamoxifen was intraperitoneally injected to the mice depending on the body weight (100 mg/kg body weight). The time points of injections are indicated in the figures. Only for the pups treated at postnatal day 1 (p1) and 2, tamoxifen was injected to the lactating mother with the same protocol[64]. For 4-week-old mice, tamoxifen was injected once per day for five consecutive days[18].

**Immunohistochemistry.** Mice were perfused with PBS and 4% paraformaldehyde (PFA). Dissected mouse brains were post fixed with 4% PFA at 4 °C overnight. Free floating brain slices (40 μm thickness) were prepared as coronal or sagittal sections using a Leica VT1000S vibratome. For immunocytochemistry, cells on coverslips were fixed with ice cold 4% PFA for 15 min. Slices or coverslips were incubated with blocking solution containing 5% horse serum and 0.5% Triton X-100 at room temperature (RT) for 1 h, followed by primary antibody incubation at 4 °C

overnight and secondary antibody incubation at RT for 2 h in blocking solution. Primary and secondary antibodies are listed in the Supplementary Tables 1 and 2, respectively.

**Magnetic cell separation (MACs) of OPCs**. MACS sorting of OPCs was performed according to the manufacturer's instruction (Miltenyi Biotec) with some modifications. Mice were perfused with cold Hank's balanced salt solution without calcium and magnesium (HBSS, H6648, Gibco) and cortices were dissected in ice cold HBSS. After the removal of debris (130-107-677, Miltenyi Biotec), cells were resuspended with 1 mL "re-expression medium" containing NeuroBrew-21 (1:50 in MACS neuro Medium) (130-093-566 and 130-093-570, Milteny Biotec) and 200 mM L-glutamine (1:100, G7513, Sigma) at 37 °C for 30 min. Cells were then incubated with Fc-receptor blocker for 10 min at 4 °C (provided with CD140 microbeads kit), followed by a 15 min incubation with 10 μL microbeads mixture containing antibodies directed against CD140 (130-101-502), NG2 (130-097-170) and O4 (130-096-670) in 1:1:1 at 4 °C.

For qRT-PCR or Western blot analysis, MACs-sorted OPCs were lysed by RIPA buffer (89900, Thermo Scientific).

**Western blot analysis**. Deeply anesthetized mice were perfused with cold PBS. Medial prefrontal cortices (mPFC) and corpus callosa (cc) were dissected in ice cold PBS. Tissues were homogenized with sucrose lysis buffer (320 mM sucrose, 10 mM Tris-HCl, 1 mM NaHCO$_3$ (pH = 7.4); 1 mM MgCl$_2$), and MACS-sorted OPCs were lysed with RIPA buffer. Both buffers were supplemented with 1 X protease inhibitors (05892970001 Roche) and 1 X phosphatase inhibitor (04906837001, Roche). Protein (5 μg) was blotted onto nitrocellulose transfer membranes (QP0907015, qpore). Membrane was blocked with 5% non-fat milk or 5% BSA (A7906, Sigma) diluted in 0.1% TBST. Primary antibodies were diluted with corresponding blocking buffer. Primary antibodies used for Western blot are listed in the Supplementary Table 3.

Secondary antibodies were: HRP anti-mouse (1:2000, A9044, Sigma) and anti-rabbit (1:2000, 111-035-045, Dianova). Membranes were illuminated with WesternBright Chemilumineszenz Substrat Quantum kit (541015, Biozym) and documented with ChemiDoc-MP. Complete and unprocessed original blots are provided in the Source Data file.

**Quantitative real time PCR**. Brain tissue or the MACs OPCs and Oli-neu cells were homogenized as described above. NucleoSpin RNA Plus XS kit (740990.50, Macherey-Nagel) was used to extract mRNA and Omniscript kit (205113, QIA-GEN) was used for reverse transcription. RT-PCR was performed using EvaGreen (27490, Axon) kit with CFX96 Real Time System (BioRad). Primer sequences for qRT-PCR are listed in Supplementary Table 4.

**Electrophysiology**. Slice preparation: Mice were anesthetized by isoflurane before decapitation, and the brain was quickly prepared and immersed in an ice-cold, oxygenated (5% CO$_2$/95% O$_2$, pH = 7.4) solution containing (in mM) 87 NaCl, 3 KCl, 25 NaHCO$_3$, 1.25 NaH$_2$PO$_4$, 3 MgCl$_2$, 0.5 CaCl$_2$, 75 sucrose and 25 glucose. Coronal or semi-sagittal slices in 300 μm thickness were prepared with a vibratome (Leica VT 1200S, Nussloch, Germany) and transferred to a nylon basket slice holder for incubation in artificial cerebral spinal fluid (ACSF) containing (in mM) 126 NaCl, 3 KCl, 25 NaHCO$_3$, 15 glucose, 1.2 NaH$_2$PO$_4$, 2 CaCl$_2$, and 2 MgCl$_2$ at 32 °C for 0.5 h. Subsequently, slices were removed from the water bath and kept at RT with continuous oxygenation prior to use.

IPSCs and EPSCs recordings of neurons: Semi-sagittal slices were transferred to the recording chamber that was continuously perfused with oxygenated ACSF containing 1 MgCl$_2$ and 2.5 CaCl$_2$ at a flow rate of 2–5 mL/min. During sEPSC recordings, 50 μM strychnine and 50 μM picrotoxin were added to block inhibitory synaptic transmission. Pyramidal neurons were identified morphologically (Axioskop 2 FS mot, Zeiss, Jena, Germany) with a 40x water immersion objective and a QuantEM 512SC camera (Photometrics, Tucson, USA). Whole-cell membrane currents were recorded by an EPC 10 USB amplifier (HEKA, Lambrecht, Germany), low pass filtered at 3 kHz and data acquisition was controlled by Patchmaster software (v2x90.5, HEKA). The patch pipettes (7–9 ΩM) were prepared from borosilicate capillaries (OD: 1.5 mm; Sutter, USA) using a Micropipette Puller (Model P-97, Sutter Instrument Co., CA). Patch pipettes were filled with the solution containing (in mM) 125 cesium gluconate, 20 tetraethylammonium (TEA), 2 MgCl$_2$, 0.5 CaCl$_2$, 1 EGTA, 10 HEPES and 5 Na$_2$ATP (pH = 7.2). Spontaneous excitatory and inhibitory postsynaptic currents (sEPSCs, sIPSCs) of pyramidal neurons in mPFC were recorded for 40 s in voltage-clamp mode at a holding potential of -70 mV and +30 mV, respectively. Currents above 10 pA were analyzed with MATLAB.

Whole-cell patch clamp recordings of OPCs: Semi-sagittal slices in 300 μm thickness were prepared from p11–14 mice. To record GABA$_A$R current in OPCs, cells were clamped at −70 mV in whole-cell mode. 100 μM GABA (Tocris) was focally applied to patched OPCs in the presence of 20 μM CNQX and 30 μM DAP5. Non-competitive picrotoxin (50 μM, Tocris) and selective GABA$_A$R antagonist SR95503 (20 μM, Tocris) were applied in the bath as well used to verify GABA$_A$R currents. To record GABAergic sPSCs of OPCs (OPC-sPSC), 20 μM CNQX (Tocris) and 30 μM DAP5 (Tocris) were added to inhibit excitatory signals.

To improve the signal to noise ratio, the Matlab tool 'Sgolay' was used. Sgolay is a Savitzky-Golay filter, which smoothes according to a quadratic polynomial that is fitted over each window of the original trace. This method is particular effective at fast variations of recorded data. After filtering, currents larger than 1 pA were selected for further analysis. In some experiments, 50 μM carbachol (Tocris) was added to stimulate inhibitory neurons.

*Data analysis*. Data generated by PatchMaster were loaded into MATLAB (Mathworks, MA, USA) with a module adapted from sigTOOL[65]. Evoked EPSC and IPSC traces from the same cells were manually checked and pooled. The average of EPSC/IPSC traces from each cell was used for analysis. Data analysis was performed using routines that were custom written in Matlab.

**EEG telemetry and analysis**. Telemetric EEG transmitter implantation was adapted from Bedner and colleagues[66]. Mice were implanted with telemetric EEG transmitters (DSI PhysioTel ETA-F10, Harvard Biosciences, Inc. Holliston, Massachusetts, USA) between 8 and 10 weeks of age. The animals were placed in a stereotactic frame (Robot stereotaxic, Neurostar, Tübingen, Germany) for implantation of depth electrodes at 3.4 mm posterior to bregma and bilaterally 1.6 mm from the sagittal suture. After post-surgical care and recovery, cages were placed on individual radio receiving plates (DSI PhysioTel RPC-1, Data Sciences International, St. Paul, USA) for synchronized EEG and video recording (MediaRecorder Software, Noldus Information Technology, Wageningen, Netherlands). EEG and video recording were performed for 24 h.

Data analysis of EEG recordings: EEG traces were analyzed with the Neuroscore software (Version 3.3.1., Data Sciences International, St. Paul, USA). After applying a general Notch filter (50 Hz band stop), relative power band values were determined with a order 13 fast-Fourier transform in 10°s epochs and later averaged per hour of recording. Statistical analysis was performed with GraphPad Prism 8.0.1.

**Behavioral analysis**. Several behavioral tests were performed with the same cohorts of mice (male and female, 9-week-old). The order of tests was arranged from low to high invasiveness to reduce the interference from the prior tests. The chamber or tested square was wiped with 75% ethanol before tests to remove odors.

*Nest building test*. Mice were moved into new single cages in the behavior testing room for habituation. After 24 h, a piece of pressed cotton in size of $7 \times 5$ cm$^2$ was placed in the cage. After 14 h, the shape and weight of the cotton was recorded and unbiasedly scored according to the criteria[67].

*Open field test*. The mice were put in the open field maze, which measured 50 cm (length) × 50 cm (width) × 38 cm (height). Mice could move and explore freely for 10 min in the open field square. In each test, the single mouse was put in the center of the square arena. The videos of these 10 min were recorded (USB webcam) and analyzed (EthoVision XT 11.5, Noldus Technology). Duration time in the center area (s), moved distance (cm) and speed (cm/s) were determined[68].

*New object recognition test*. The mice could freely explore the two identical objects placed at a distance of 8.5 cm from the side walls in two opposite corners of the apparatus for 10 min. After 40 min, one of the objects was replaced by a novel one, and the test mouse was allowed to explore again. Preference index was defined as the percentage of the time exploring one identical object within the total time exploring both objects. Recognition index was defined as the percentage of the time exploring the novel object among the total time of exploring both objects.

Three chamber social behavior test: The three-chamber box was employed for social behavioral studies[69]. Three chambers were equally sized and separated by two walls evenly distributed in the box. A square door at the bottom center of each door allowed free running of the mice within the three chambers. Two empty wire cages were placed at the side chambers, leaving the center chamber empty. For habituation, mice were kept in the center chamber for 10 min, followed by a 10 min-habituation session with access to all three chambers. For the sociability test, the test mouse was placed in the center chamber, while a mouse of similar age was kept under the stainless wire cage in one of the side chambers. The other chamber contained an empty wire cage. For 10 min, the test mouse could select freely all three chambers. For the social novelty test, a novel mouse was placed under the other empty cage being an unfamiliar mouse, and the prior mouse was familiar one. The test mouse was again allowed to freely explore both animals for 10 min. The experiment was video-recorded and the time that the test mouse spent in each chamber and the time of sniffing was analyzed.

**Cell culture**

*Cell line Oli-neu*. The murine oligodendroglial precursor cell line Oli-neu[70] was kindly provided by Professor Jacqueline Trotter (University of Mainz). Undifferentiated Oli-neu cells were incubated at 37 °C and 5% CO$_2$ in poly-L-lysine (Merck) coated cell culture flasks (Greiner Bio one) for expansion or cell culture dishes (Greiner Bio one) for experiments. Sato medium consisting of DMEM high glucose medium (Fisher Scientific) with supplementation of 10 μg/ml transferrin (Sigma), 10 μg/ml insulin (Santa Cruz), 100 μM putrescine (Sigma), 200 nM progesterone (Sigma), 500 nM tri-iodo-thyrodine (Sigma), 220 nM sodium selenite (Sigma), 520 nM L-thyroxine (Sigma) and 1.5% normal horse serum (Fisher

Scientific) was used for culturing and proliferation of cells. For each independent experiments, $2 \times 10^5$ cells from the same passage were seeded in a 60 mm Petri dish. 48 h later, the medium was changed to fresh, with or without (2S)-3-[[(1S)-1-(3,4-Dichlorophenyl)ethyl]amino-2-hydroxypropyl](phenylmethyl)phosphinic acid (CGP 55845, 50 μM, Tocris) for 24 h. Cells were lysed with Qiazol (Qiagen) for qRT-PCR in a blind manner. Conditional medium of control cells was collected for further neuronal treatment. In total, three independent experiments were performed.

*Primary culture of cortical neurons.* Cortical neurons were isolated from p0 pups (C57BL/6N mouse strain). Briefly, the cortex was dissected from the whole brain in ice cold Earle's Balanced Salt Solution (EBSS, Gibco). The cortex was then digested with 35 units papain (Worthington, NJ) for 45 min at 37 °C, followed by gentle mechanical trituration. $1 \times 10^5$ cells were seeded on 25 mm glass coverslips in 24-well culture plates for further immunostaining. Each mouse brain was independently prepared for one 24-well plate, considered as one independent experiment. The glass coverslips were pre-coated with a mixture of coating solution containing 17 mM acetic acid, poly-D-Lysine (Sigma, P6407) and collagen I (Gibco, A1048301). Neurons were cultured in NBA culture medium that contained 10% FCS, 1% penicillin-streptomycin, 1% GlutaMAX and 2% B-27 supplement (Gibco) for 7d at 37 °C with 5% $CO_2$ before the experiment. To examine the effect of TWEAK on neurons, conditioned Oli-neu culture medium supernatant or Sato, was applied to neuron NBA culture medium, with a ratio of 1:2 to NBA resulting in 1 ml medium per 24-well and 2 ml medium per 6-well. Cells were collected after 6 h with different culture condition with or without 20 μM TWEAKR inhibitor L524-0366 (509374, Calbiochem) for further analysis.

**Image acquisition and analysis.** Brain slices were scanned with the fully automated slide scanner AxioScan.Z1 (Zeiss, Jena) and LSM 710 and 780 confocal microscope (Zeiss, Jena). Cell counting was manually performed by using ZEN software (Zeiss, Jena). The lengths of nodes and paranodes were measured using the 'straight' tool of Fiji software. The Imaris (version 9.6) with surface tracking functions of volume analysis and vesicle classification were used for the analysis of MOG (Fig. 2d, e, h) and vGAT immunostainings (Fig. 3h–k, Supplementary Fig. 8b–d). Briefly, for the MOG/PV/SMI312 immunolabel analysis, after background subtraction a surface of the axonal SMI312 immunostaining was rendered and respective volumes estimated. All the immunoreactive fragments, of which the volume was larger than 0.6 μm³, were selected for analysis. To specifically select the axons of PV⁺ interneurons, a filter with 'Minimum intensity' of the PV channel was applied and only double positive fragments were selected for volume analysis. On this surface, an additional filter 'Minimum intensity' of the MOG channel was applied to quantify the myelination of PV⁺ interneurons. For the myelination of all axons (inhibitory and excitatory), the MOG channel filter was directly applied to the SMI 312 surface. For the vGAT analysis, background subtraction was followed by 'Surface' mode in Imaris. On this surface, the distance between vGAT puncta and OPCs was analyzed and classified in '<200 nm' and '>200 nm', as putative presynapses on OPCs or on other cells, respectively. The volume of each vGAT punctum and total number of vGAT puncta were analyzed by Imaris. Cell counting and Imaris analysis were carried out in a blind manner.

**Bromodeoxyuridine assay.** Adult animals received 1 mg/ml bromodeoxyuridine (BrdU) (B5002, Sigma-Aldrich, St. Louis, MO) dissolved in drinking water for seven consecutive days. For juvenile mice, single shots of 10 mg/ml BrdU dissolved in 0.9% NaCl were intraperitoneally injected to the p13 pups and analyzed at p14.

**Statistics and reproducibility.** The statistical analyses of all data were performed with GraphPad Prism 8.0.1. For all immunostainings, four brain hemispheres from randomly selected brain slices of each mouse were studied. In addition, for the analysis of vGAT and PV/SMI312/MOG, at least 6 OPCs or 8 regions of interests per mouse was analyzed, respectively. For the in vitro study, three independent primary cell preparations or 6 independent experiments of the Oli-neu cell line were performed. Within one independent experiment, cells from the same passage (Oli-neu cells) or the same preparation (primary neurons) were randomly distributed to different experimental groups prior to the treatment. In the PV/CC-3 immunostaining experiments, each group had two replicates. Prior to statistical analysis, data were tested for normal distribution with the Anderson-Darling test and outlier identification using the Rout method (Q = 1%). Outliers were excluded from the statistical analysis. For the normally distributed dataset, unpaired *t*-tests, paired *t*-test (for studies of behavior), one-way ANOVA and two-way ANOVA were used (indicated in each figure legend), while the Mann–Whitney test was used for non-normally distributed datasets. *P*-values are indicated in the figures and legends. For the in vivo experiments, each data point represents the data obtained from a single mouse (except for electrophysiology and EEG recordings). The total mouse numbers are given in the figure legends. For electrophysiology and EEG recordings, each data point refers to a single cell or single recording unit (hour), respectively, and the numbers of cells and mice are given in the figure legends. Data are shown as mean ± SEM, except for the violin plots (Figs. 3k and 6c), where the median and quartiles are indicated as thick and thin dashed lines, respectively.

**Reporting summary.** Further information on research design is available in the Nature Research Reporting Summary linked to this article.

## Data availability
Source data are provided with this paper. Further data to support the findings can be obtained upon request to the corresponding authors.

## Code availability
Custom codes used for electrophysiology analysis are available at GitHub[71] (https://github.com/XianshuBai/OPC-GABABR-interneuron).

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

## Acknowledgements

We thank Daniel Schauenburg for excellent animal husbandry and Frank Rhode, Alexander Grissmer and Samantha Bechet for experimental assistance, Qinghai Tian for technical support. We are also grateful to Prof. Jacqueline Trotter (University of Mainz, Mainz, Germany) for providing the Oli-neu cell line and Prof. Leda Dimou (University of Ulm, Ulm, Germany) for sharing MACs sorting protocols of OPCs. This work was supported by grants from the Deutsche Forschungsgemeinschaft (SPP 1757, DFG-237267101; SFB 894, DFG-157660137 to F.K.; FOR2289, DFG-262890264 to F.K. and A.S.; SPP1757 Young Investigator grant to X.B.), the BMBF (EraNet-Neuron, JTC2014: BrIE to F.K.), the European Commission (H2020-MSCA-ITN-2016 EU-GliaPhD #722053 to F.K.), the Romanian UEFISCDI (PCE 227; PN-III-P4-ID-PCE-2020-2477 to F.K.) and University of Saarland (HOMFORexzellent2017 and NanoBioMed Young Investigator grant 2021 to X.B.).

## Author contributions

X.B. initiated and designed the project. L.F., X.B., N.Z., L.C., H.C., N.H., C.L. and A.S. performed experiments. L.F., X.B., N.Z., L.C. and R.Z. analyzed data. B.B., W.H., and C.M. provided materials. X.B. and F.K. supervised the study and wrote the manuscript with comments of the other authors.

## Funding

## Competing interests

The authors declare no competing interests.

**Additional information**

