## [Peer review file · Nature Communications]

REVIEWER COMMENTS

Reviewer #1 (Remarks to the Author):

In this manuscript, Fang et al. investigate communication between OPCs and inhibitory interneurons during development. The group discovers a novel function of OPCs: the control of inhibitory interneuron number in the mPFC. The study is also novel because this function depends on a neurotransmitter receptor located on OPCs, the metabotropic GABA-B receptor. The manuscript is intriguing and important. Learning that OPCs and neuronal subtypes communicate to coordinate appropriate cell number and myelination provides key insight into how neuronal subtype-specific myelination could arise in the CNS.

My suggestions are mainly directed at strengthening the assertion that interneuron activity (and not number) underlies the myelination deficit and determining the extent and cell autonomy of the myelination deficit.

1. The authors find that *gabbr1* deletion in OPCs is associated with both an excess of interneurons and an apparent decrease in interneuron activity, as well as a deficit of myelination. It should not be inferred that abnormal myelination results from decreased activity in this model without an independent method of either suppressing interneuron activity (in which OPC *gabbr1*, and interneuron number, are both wildtype/normal) or increasing the number of interneurons (in which interneuron activity, and OPC *gabbr1*, are both wildtype/normal). There are too many confounding variables to conclude that activity is responsible for the myelin decrease without testing activity and/or interneuron number independently.

2. "Myelin volume", as assessed (Figure 1n) is MBP immunoreactivity volume and is not the same thing as the volume of myelin. As the authors note, MBP protein content can vary on different axon types, and the MBP mutant mouse (*shiverer*) still forms myelin, albeit uncompact, so MBP immunoreactivity is not equivalent to myelin volume. The authors should use a method that labels myelin, like fluoromyelin or potentially staining for a myelin protein with unperturbed expression (like PLP, which the group found to be unaffected in this manipulation) to make claims about whether myelin volume changes in *gabbr1* cko.

3. How many differentiated oligodendrocytes are *gabbr1* cko? The authors quantify recombination efficiency in OPCs and find roughly 85% recombination. They also show that *gabbr1* cko OPCs have impaired differentiation. Therefore, one might think the population of differentiated oligodendrocytes would be overrepresented with *gabbr1* wildtype cells that were able to differentiate unimpeded. Assessing whether oligodendrocytes are predominantly *gabbr1* wildtype or cko would help with

interpretation of the myelination deficit (ie, is it cell autonomous or not?). Please also see minor point #3 below.

4. The labeling of “20 nm vesicles” needs to be rephrased, because 20 nm was not measured. I understand the reasoning for this label, based on assumed synaptic cleft distance, but this is a misleading label because the distance wasn't and couldn't be (with diffraction-limited microscopy) measured. Perhaps the groups could be labeled something like "putative OPC synapse" and "other synapse" for the current <20 nm and >20 nm groups, respectively.

5. Line 973 (Statistics): Data were tested for outliers and normality but the authors do not state what these tests were or how the outcomes of these tests informed downstream analysis. For example, were outliers removed? Were non-normally distributed samples tested differently? Additionally, throughout the paper it is unclear in many of the figure legends what n is referring to. The Methods state that n is animals, but this is not clear in experiment types like Supplemental Fig 5c in which stained cells are shown and “n=4” (how many cells, sections, animals did these come from?).

6. Is TWEAKR only expressed by inhibitory interneurons?

Minor suggestions

1. The authors mention several times that interneuron “peak apoptosis” occurs at P7, so it is unclear to me why the choice was made to wait until after this peak to induce *gabbr1* excision (P7-P8). Might a more exacerbated phenotype arise if deletion took place sooner, potentially causing a greater surplus of interneurons to survive?

2. I suggest that the findings of Swire et al. (2019, eLife) and Makinodan et al. (2012, Science) be considered in the interpretation of Figure 4. Especially provided the myelination deficit in these animals, social impairment could reasonably follow mPFC hypomyelination and these previous studies provide context in this area.

3. Knowing whether oligodendrocytes lack *gabbr1* would be helpful in interpreting the impaired myelination data (is there an autonomous effect of *gabbr1* in promoting myelination, for example). In Supp Fig 3, a PLP-cre strategy is used to delete *gabbr1* in oligodendrocytes and then oligodendrocyte population (cell number) is assessed. Determining if these animals have normal myelin would also be very impactful for the field, especially given the unclear role of NT receptors in oligodendrocyte function.

4. Line 131: "Required by their fast-spiking activity, parvalbumin (PV)+ interneurons are the most abundantly myelinated interneurons". It is known that these interneurons are abundantly myelinated but it is a stretch (to my knowledge) to assert that this is specifically because of their firing rate.

5. Line 199: "Obviously, OPCs receive GABAergic input through GABABR and send back pro-apoptotic signals to interneurons." I see that this is the model suggested by the data, but please consider dropping "obviously".

6. Line 337: "MBP is preferentially expressed in the myelin of inhibitory axons, while PLP is more prominent in excitatory axons PLP." This is not supported by the reference and should be corrected. Micheva et al. (2016) did find a slight enrichment of MBP in myelin around inhibitory neurons, but there was no such bias for PLP on any axon type.

7. Supp Fig 2c: axis label typo, paranode "lengtha".

8. Supp Fig 10c: the difference between baclofen and baclofen + CGP is not explicitly tested, and the statistical test used for the other pairwise comparison is absent from the legend.

9. Supp Fig 11c,d: Were animals tested for different total amounts of time? It would seem the sniffing time amounts (F/UF) should add up to roughly the same amount of time per mouse, but they vary wildly.

10. Line 171: "...including a smaller volume of vGAT (Fig. 2h) and less puncta (Fig. 2i) at the OPC surface in cKO mPFC, which can explain the attenuated firing rate of the interneurons in the adult." It isn't clear to me what the authors mean here.

11. Line 742: Behavior was only performed in male mice. A group containing only males is not representative. Including some female mice is advised to increase the representativeness and applicability of the behavior results in this manuscript.

12. Line 921: I assume you mean 1×10^5 cells and the exponent got dropped.

13. Line 957: Is "ia" supposed to be "ie"?

Reviewer #2 (Remarks to the Author):

Authors analyzed the impact of conditional KO of GABA_bR from OPC in mice and demonstrated that such a manipulation impacts the differentiation of PFC OPC maturation, and also PFC PV interneuron myelination and their density (inhibiting apoptosis). They also found reduction of inhibitory inputs onto PFC pyramidal neurons and deficits in social and object novelty behavior tasks. The authors also demonstrated an in vitro evidence that PV apoptosis is regulated by OPC derived TWEAK signaling under the regulation of GABA_bR. Overall, I found the findings are interesting, and are consistent with the recent findings demonstrating a bi-directional link between PV interneuron and OPC through GABA signaling onto OPC (Benamer et al Nat Communications 2020). It adds new insights that not only GABA_aR but also GABA_bR plays a key role in this process, and also provide some novel mechanistic insights into how GABA signaling in OPC regulates PV apoptosis. However, to more convincingly support some of authors' claims, I suggest to more thoroughly characterize 1) GABA_bR deletion from OPC across development, 2) GABA_bR current changes in OPC, 3) myelination of excitatory neurons, 4) PV cell functional properties. Given that findings on TEWAK pathway is limited to in vitro setting, additional in vivo characterization and discussion on the mechanism will also strengthen the manuscript.

Major comments:

#1 Throughout the manuscript, conditional knock-down of OPC-GABA_bR needs to be better validated across development. It is important to show the timeline of deletion of GABA_bR for some of experiments when analysis is conducted at early ages such as p5-7, p10-14 (data similar to Fig1c and d at earlier age). Even if tdTomato is positive, it does not necessary mean GABA_bR per se is successfully knocked-out.

#2 It would be helpful to show the extent of both GABA_a and GABA_b receptor mediated currents in developing OPC, and how knock-out of GABA_bR in OPC impacts these currents and membrane potential (given that recent publication of GABA_aR knock-out from OPC showed similar results, Benamer et al Nat Communications 2020).

#3 Selectivity of deficits in intracortical myelination of PV interneurons is not well supported by the presented data. It is more convincing if analysis is extended to myelinated axons of excitatory neurons within cortex (such as thalamo-cortical projections) to make such a claim.

#4 line 157, 247-8, 289: Hypoactivity of PV interneurons not well supported. Only data indirectly related is reduced sIPSC frequency from pyramidal neurons. Direct electrophysiological characterization of PV

cells (e.g. excitability) in slice, or in vivo activity (e.g. optogenetic tagged recording or early immediate gene expression in PV) will help to support this claim.

#5 In vivo evidence of the role of TWEAK is limited. I suggest to at least show the expression pattern of TWEAKR especially in PV cells in tissue by immunohistochemistry. As all associated data is from cultured condition, limitation of the findings (lack of in vivo study) also needs to be discussed in the discussion section.

#6 One of the limitations of the current study is the lack of mechanism of GABAbR regulation of TWEAK secretion. Can author discuss this more?

Minor comments

Is there any PV density difference in different layers of cortex in mPFC? (Fig2)

Fig4b: Is the power difference selective to gamma band? Please show data of other power band.

Throughout the manuscript, it is unclear if the sample number is cell, section, or animal. Please provide the information explicitly.

line 256. My understanding is the new object recognition is not PFC dependent.

line 275-6: As manipulation is not selective at PFC, link between PFC and behavior is not established by this study.

It would be helpful to discuss similarities and differences between the action of GABAaR and GABAbR in OPCs to PV interneurons (especially findings reported in Benamer et al 2020).

Reviewer #3 (Remarks to the Author):

In the manuscript “Impaired bidirectional communication between interneurons and oligodendrocyte precursor cells affects cognitive behavior”, Fang et al present exciting findings providing new insights on the roles of early communication between interneurons and OPCs in mPFC myelination, interneuron activity, cortical network function and cognitive behavior. Specifically, the authors show that interruption of GABAB receptor-mediated signaling to OPCs (via deletion of *gabbr1* subunit of GABABRs selectively in OPCs) at the end of the first postnatal week results in impaired mPFC myelination and reduced inhibitory network activity, leading to deficits in cognition and social behavior. Besides they revealed that the release of the TNF-related cytokine TWEAK by OPCs, in a GABABR mediated manner, induces neuronal apoptosis to regulate the density of parvalbumin (PV) interneurons.

Therefore, the study of Fang et al. sheds new light on the role of bidirectional communication between interneurons and OPCs in the regulation of cell densities and myelination. The study combines various approaches, from patch-clamp recordings to testing behavior of animals that makes the conclusions solid and of significant interest to the broad fields of activity-dependent myelination and cortical inhibitory circuits. Overall, the study significantly contributes to understand the role of interneuron-OPC interactions. However, some conclusions are confusing and some clarifications will be helpful:

1. In Figure 1, the authors observed a decrease of mature oligodendrocytes (OL) associated to a hypomyelination in the mPFC of the KO compared to the Control while they found no differences in corpus callosum. Did they also observe these effects in other cortical area, or is it specific of the mPFC?

2. The authors observed a decrease in mature OL in the mPCF KO without affecting the density of OPCs and conclude that OPC-GABABR play a role in OPC differentiation. However, the fact that the pool of OPCs is not affected may suggest that there is an OL cell death increase that causes the difference in OL cell number rather than less differentiation. This point needs more explanation by looking at Caspase-3 expression in OL, for instance.

3. Besides, they observed in the KO mPFC an increase in the paranodal length without affecting the node length. In Figure 1j, the authors show differences in the distribution of the paranodes length, but is there also a significant difference of their mean size. The authors should also discuss more how this could affect PV-interneuron function and properties. For instance, could longer paranode affect protein clustering and potassium channel expression (Kv) in the juxtaparanode region?

4. In Figure 1m-n, the authors report in the KO a decrease in the volume of myelin sheaths wrapping PV interneurons by doing only a PV and MBP double immunostaining. However, in figure 1m showing 3D reconstructions of myelinated PV+ axons, we see less PV+ processes in KO compared to Control mice. The reported differences could thus be due to less PV+ axons rather than less myelinated PV+ axons. Did the authors normalize the quantification of myelinated PV+ axons to the volume of PV+ axons? Could the authors give more details about how the quantification was done? To strengthen the conclusion that

GABABR of OPCs are involved specifically in myelination of inhibitory axons, the authors should stain specifically PV+ axons by using an axonal marker (SMI-312 for instance) in addition to PV and MBP double immunostaining, and normalized to the total volume of PV+ axons. Alternatively, they can quantify the co-localization of PV and MBP specifically in biocytin loaded PV-interneurons.

5. In Figure 2b, the authors show an increase of PV interneuron density in the KO mPFC. What about the density of PV-interneurons among layers? Do they observe these differences if they compare the densities between upper and deeper layers, specifically? This is relevant in this study since they studied neuronal activity in layer 5.

6. The authors quantified synaptic vGAT density of OPC surface and observed a decrease in the density of vesicles on OPCs but also on other cells in the KO. First, this effect is not specific of OPCs and the authors should discuss how a loss of GABABR signaling to OPCs could generate a global decrease in puncta densities. Besides, the conclusion that less GABAergic vesicles on OPC surface could explain the attenuation of firing rate in interneurons (line 171) based on those results is not correct and should be modified. What is the rationale for quantifying the density of vesicular GABA transporter close to OPCs as a readout of interneuron activity? To correlate the decrease of vesicle densities and neuronal GABAergic activity, the authors should quantify miniature inhibitory postsynaptic currents in neurons. The authors also conclude from these results that interneurons exhibit lower activity and transmit less GABAergic signals to OPCs. Again, this assumption is not correct and the author should quantify directly GABAergic activity in OPCs by patch-clamp recordings to reach this conclusion.

Point-to-point response

Reviewer #1 (Remarks to the Author):

In this manuscript, Fang et al. investigate communication between OPCs and inhibitory interneurons during development. The group discovers a novel function of OPCs: the control of inhibitory interneuron number in the mPFC. The study is also novel because this function depends on a neurotransmitter receptor located on OPCs, the metabotropic GABA-B receptor. The manuscript is intriguing and important. Learning that OPCs and neuronal subtypes communicate to coordinate appropriate cell number and myelination provides key insight into how neuronal subtype-specific myelination could arise in the CNS.

My suggestions are mainly directed at strengthening the assertion that interneuron activity (and not number) underlies the myelination deficit and determining the extent and cell autonomy of the myelination deficit.

1. The authors find that *gabbr1* deletion in OPCs is associated with both an excess of interneurons and an apparent decrease in interneuron activity, as well as a deficit of myelination. It should not be inferred that abnormal myelination results from decreased activity in this model without an independent method of either suppressing interneuron activity (in which OPC *gabbr1*, and interneuron number, are both wildtype/normal) or increasing the number of interneurons (in which interneuron activity, and OPC *gabbr1*, are both wildtype/normal). There are too many confounding variables to conclude that activity is responsible for the myelin decrease without testing activity and/or interneuron number independently.

We thank the reviewer for this remark. Indeed, we do agree that our results do not support the conclusion that perturbed myelination in our mouse model is attributed to the impaired interneuronal activity. Our conclusion was based on the observation that at p10, prior to myelination onset (about p14, Supplementary Fig. 7b, c), interneuron activity was already suppressed (Fig. 3g-n), ahead of the reduced OPC differentiation and MBP mRNA expression at p14 (Fig. 4). In the revised manuscript, besides the inadequate activity of interneurons in the cKO mPFC, we discuss also the morphological contributions to the hypomyelination including the changes in interneuron axons and paranode structures.

Page 17, line 436: *'Additionally, a potential change in the PV interneuron morphology is detected since the relative axon volumes of PV interneurons in cKO and ctrl mice were identical despite a larger population of PV⁺ interneurons in the mPFC of mutant mice, e.g. thinner axonal caliber, shortened axons*

or less arborized axons in the cKO mPFC. However, such a detailed anatomical analysis is beyond the scope of this study.'

Page 18-19, line 466: 'The elongation of paranode will be accompanied with redistribution of Kv1 channels from under juxtaparanode towards paranode in the brain of multiple sclerosis and aging⁵⁶. These exposed Kv1 channels are more active. As a result, these changes hinder the action potential propagation, which in turn reduces the synaptic communication.'

2. "Myelin volume", as assessed (Figure 1n) is MBP immunoreactivity volume and is not the same thing as the volume of myelin. As the authors note, MBP protein content can vary on different axon types, and the MBP mutant mouse (shiverer) still forms myelin, albeit uncompact, so MBP immunoreactivity is not equivalent to myelin volume. The authors should use a method that labels myelin, like fluoromyelin or potentially staining for a myelin protein with unperturbed expression (like PLP, which the group found to be unaffected in this manipulation) to make claims about whether myelin volume changes in gabbr1 cko.

We agree with the reviewer that myelin volume should not be based on MBP immunostaining, especially when MBP is downregulated in the cKO mPFC. We have repeated the immunostaining using PV, SMI 312 (pan axonal marker) and the transmembrane myelin oligodendrocyte glycoprotein MOG, of which gene expression was unaffected by the OPC-GABA_BR deletion (**Supplementary Fig. 4g**). We quantified the volume of MOG⁺ myelin sheaths wrapped around PV⁺ axons as well as of all axons. Our results showed that the relative volume of MOG⁺ myelin sheaths of PV axons was decreased in the cKO mPFC compared to the controls, while that of all axons remained unchanged (**Fig. 2d, e**). These data strongly suggest that the myelination of PV interneurons in the cKO mPFC is indeed deteriorated.

3. How many differentiated oligodendrocytes are gabbr1 cko? The authors quantify recombination efficiency in OPCs and find roughly 85% recombination. They also show that gabbr1 cko OPCs have impaired differentiation. Therefore, one might think the population of differentiated oligodendrocytes would be overrepresented with gabbr1 wildtype cells that were able to differentiate unimpeded. Assessing whether oligodendrocytes are predominantly gabbr1 wildtype or cko would help with interpretation of the myelination deficit (ie, is it cell autonomous or not?). Please also see minor point #3 below.

We thank the reviewer for the suggestion. To investigate whether or not the composition of wildtype and cKO oligodendrocyte are as same as in the ctl situation, we quantified the proportion of $CC1^+tdT^+$ cells over all $CC1^+$ cells at 2w and 9w (**Fig. 4j-l**), when the oligodendrocyte density was already reduced. Our results showed that the population of recombined oligodendrocytes among all oligodendrocytes were unchanged in the cKO mPFC at both time points. In addition, by analyzing the density of wildtype and cKO oligodendrocytes, we did not observe significant overrepresentation of wildtype cells. Nevertheless, the density of wildtype cells remained similar while that of recombined oligodendrocytes was drastically reduced in the cKO brain. These observations indicate that the impeded oligodendrogenesis cannot be completely attributed to $GABA_B$ mediated differentiation, rather to the impaired interneuron activity. Also the late induction of gene ablation at 4w did not reduce OPC differentiation (**Supplementary Fig. 10**). The reduction of OPC differentiation is likely due to a suppressed interneuron activity in the cKO brain, but is probably mediated by a different pathway. This part has been implemented to the 'Results'.

Page 11, line 261: 'These data strongly suggested that the OPC- $GABA_B$ was not directly involved in OPC differentiation, at least during early postnatal weeks. In addition, the percentage of recombined tdT^+ OLs of all OLs in both cKO and ctl mPFC did not differ at 2w and 9w (Fig. 4i-k), despite decreased densities of mature $CC1^+$ OLs (Fig. 4c) or $CC1^+tdT^+$ OLs (Fig. 4l) in the cKO mPFC. Thereby, it implied that the reduced OPC differentiation during development was rather due to the impaired interneuron-OPC communication.'

4. The labeling of "20 nm vesicles" needs to be rephrased, because 20 nm was not measured. I understand the reasoning for this label, based on assumed synaptic cleft distance, but this is a misleading label because the distance wasn't and couldn't be (with diffraction-limited microscopy) measured. Perhaps the groups could be labeled something like "putative OPC synapse" and "other synapse" for the current <20 nm and >20 nm groups, respectively.

We thank the reviewer for pointing out the misleading term "< or >20 nm", where we simply applied the size of the synaptic cleft as the distance between the rendered surfaces generated by the Imaris software. Now, we re-analyzed the surface-rendered images and applied a more reasonable distance in respect to the diffraction limits. Based on super-resolution nanoscopy of inhibitory synapses (Crosby et al., 2019), we quantified the vGAT immunoreactivity of vGAT close to the OPC surface by defining 200 nm as the new threshold (**Fig. 3 h-k; Supplementary Fig. 8b-d**), and refer to vesicles released to a 'putative OPC synapse' or 'other synapse'.

5. **Line 973 (Statistics):** Data were tested for outliers and normality but the authors do not state what these tests were or how the outcomes of these tests informed downstream analysis. For example, were outliers removed? Were non-normally distributed samples tested differently? Additionally, throughout the paper it is unclear in many of the figure legends what n is referring to. The Methods state that n is animals, but this is not clear in experiment types like Supplemental Fig 5c in which stained cells are shown and “n=4” (how many cells, sections, animals did these come from?).

We apologize that our statistical evaluations were not presented clear enough. In the revised version, we have clarified the procedures of data analysis in the ‘Statistics’ in the ‘Materials and Methods’ section. Briefly, prior to the statistical analysis, we evaluated the data for outliers and normal distribution with GraphPad 8.0.1. The outliers were excluded from the analysis, normally distributed data were analyzed with t-tests (paired or unpaired as indicated in the figure legends) while for the non-normally distributed data, Whitney Mann tests or Wilcoxon tests were employed (also indicated in the figure legends). Now, we provide sample sizes and the meaning of ‘n’ in the ‘statistics’ as well as in each figure legend. In most of the figures, each single dot indicates data obtained from a single mouse (n=mouse number), and we have corrected the mouse number and cell numbers in all figure legends in the revised manuscript.

6. **Is TWEAKR only expressed by inhibitory interneurons?**

*According to the RNA sequencing study (Zhang et al., 2014); <http://www.brainrnaseq.org/>) and other studies (Cheadle et al., 2020), TWEAKR mRNA is robustly detected in microglia and endothelial cells. The roles of TWEAKRs for microglia and endothelial cell functions are unknown yet. In addition to the immunohistochemical detection of FN14/TWEAKR in interneurons of the mPFC or in cell culture, we can also functionally demonstrate TWEAKR expression employing an in vitro system without microglia and endothelial cells. We found that a TWEAKR antagonist could rescue the interneuron apoptosis induced by a conditioned medium containing TWEAK (**Fig. 5f-i**). Therefore, we feel confident to state that OPC-derived TWEAK indeed acts on interneuronal TWEAKRs and transduces apoptotic signals. We also addressed this point more clearly in the ‘Discussion’.*

Page 17, line 420: *‘As shown by RNA sequencing, microglia express rather high levels of TWEAKR during development in the cortex⁵⁰. However, whether or not a TWEAK-based OPC-microglial crosstalk contributes to the elimination of interneurons during development is not clear yet. At this point, we can not exclude a contribution of TWEAKR-expressing microglia, which requires future experiments. So far,*

our in vitro data show that in a pure culture system, TWEAKR antagonist can interfere with interneuron apoptosis induced by TWEAK from conditioned medium of OPCs (Oli-neu cell line).'

Minor suggestions

1. The authors mention several times that interneuron “peak apoptosis” occurs at P7, so it is unclear to me why the choice was made to wait until after this peak to induce gabbr1 excision (P7-P8). Might a more exacerbated phenotype arise if deletion took place sooner, potentially causing a greater surplus of interneurons to survive?

We apologize for not explaining this point clearly enough. First of all, due to the early postnatal proliferation rate of OPCs, a tamoxifen-induced recombination at p1-2 will affect only 40-60 % of OPCs at p5-7 (Supplementary Fig. 12) while the induction of recombination at p7-8 results in 70-80 % of recombined cells at later time points (Fig. 1c, Supplementary Fig. 7e-h). In addition, as the reporter recombination efficiency showed (% of Pa^+tdT^+/Pa^+), the recombination efficiency drops at p7 compared with p5 (Supplementary Fig. 12), which is highly likely due to the fast turnover of OPCs during first postnatal week (elimination of first two waves OPCs and newly generated OPCs (Kessaris et al., 2006)). We also cannot exclude that a certain percentage of the low recombination rate should be attributed to lower levels of effective tamoxifen metabolites in the pups, since tamoxifen was administered to the lactating mother for the p1-2 protocol, while it was directly injected into the pups at p7-8. Since the rate of interneuron apoptosis increases drastically postnatally and reaches a peak at p7, from which it stays rather high till p11 (Southwell et al., 2012), we decided to employ the p7-8 protocol for the majority of the experiments, which still impeded interneuron apoptosis about 50 % at p10 and p14 (Supplementary Fig. 11).

2. I suggest that the findings of Swire et al. (2019, eLife) and Makinodan et al. (2012, Science) be considered in the interpretation of Figure 4. Especially provided the myelination deficit in these animals, social impairment could reasonably follow mPFC hypomyelination and these previous studies provide context in this area.

We thank the reviewer for this suggestion. We included the two studies to discuss our behavioral observations.

Page 19, line 482: *'In early socially isolated mice the observed cognitive impairment is also accompanied by a strong hypomyelination in the mPFC^{57,58}. A proper E/I ratio in the prefrontal cortex,*

especially the E/I balance in the postnatally developing mPFC, is extremely important for social cognition
37, 59 ,

3. Knowing whether oligodendrocytes lack gabbr1 would be helpful in interpreting the impaired myelination data (is there an autonomous effect of gabbr1 in promoting myelination, for example). In Supp Fig 3, a PLP-cre strategy is used to delete gabbr1 in oligodendrocytes and then oligodendrocyte population (cell number) is assessed. Determining if these animals have normal myelin would also be very impactful for the field, especially given the unclear role of NT receptors in oligodendrocyte function.

*This is a very important point and very intriguing to understand whether GABA_BRs of mature oligodendrocytes have an impact on myelination. In the revised manuscript, we evaluated the levels of MBP expression in the mPFC and corpus callosum of the OL-GABA_BR-deficient mice by Western blot. To be consistent with the ablation in OPCs, we injected tamoxifen at p7-8 in the PLP-CreERT2 mice in addition to the injection at 4 weeks of age. In both brain regions the levels of MBP were comparable in control and mutant mice (**Supplementary Fig. 6g**). Since we also did not observe changes in oligodendrocyte differentiation, we did not address structural alterations of myelin further. This has to remain for a future and more detailed analysis. Our data strongly suggest that the impeded interneuron elimination and impaired activity (p10-14) is mainly attributed to the deletion of GABA_BRs from OPCs. At p10 only very few recombined oligodendrocytes per brain slice can be detected (**Fig. 4c**). Rather we observe a reduced oligodendrocyte density subsequently to the detection of interneuronal hypoactivity at p14 (**Fig. 3h-n, Fig. 4c**).*

4. Line 131: “Required by their fast-spiking activity, parvalbumin (PV)+ interneurons are the most abundantly myelinated interneurons”. It is known that these interneurons are abundantly myelinated but it is a stretch (to my knowledge) to assert that this is specifically because of their firing rate.

We are grateful for the comments. We have revised the sentence in the manuscript.

5. Line 199: “Obviously, OPCs receive GABAergic input through GABABR and send back pro-apoptotic signals to interneurons.” I see that this is the model suggested by the data, but please consider dropping “obviously”.

We have rephrased the sentence in the revised manuscript.

6. Line 337: “MBP is preferentially expressed in the myelin of inhibitory axons, while PLP is more prominent in excitatory axons PLP.” This is not supported by the reference and should be corrected. Micheva et al. (2016) did find a slight enrichment of MBP in myelin around inhibitory neurons, but there was no such bias for PLP on any axon type.

We thank the reviewer for the comment. What we wanted to express was that MBP is more pronounced in the interneuronal myelin sheath while PLP is not. Our previous description was ambiguous. We have rephrased this part in page 19.

Page 19, line 474: Myelin proteins of inhibitory and excitatory axons are differently expressed, e.g. MBP is preferentially expressed in the myelin of inhibitory axons¹⁶.

7. Supp Fig 2c: axis label typo, paranode “lengtha”.

We apologize for our typo and corrected it in the **Supplementary Fig. 4b**.

8. Supp Fig 10c: the difference between baclofen and baclofen + CGP is not explicitly tested, and the statistical test used for the other pairwise comparison is absent from the legend.

We apologize for the carelessness. The comparison between Baclofen and Baclofen + CGP is added in the **Figure for reviewer**.

Here, we would like to point out that in the revised manuscript, we replaced the Western blot results of TWEAK with qRT-PCR after careful consideration on the specificity of TWEAK antibody. From the information supplied by the manufacturer, the antibody is supposed to detect the band only at 36 and/or 30 kDa depending on the tissue type (https://www.novusbio.com/products/tweak-tnfsf12-antibody_nbp1-76695). However, we observed multiple bands ranging from 25-50 kDa with the strongest immunoreactivity at 48-50 kDa. These might be due to the glycosylation of TWEAK protein in

the cell line Oli-neu. Therefore, alternatively, we performed qRT-PCR of Oli-neu cells treated with or without CGP 55845 (20 μ M) (Fig. 5d). Our results showed that the TWEAK mRNA level was reduced about 50 % after blocking the GABA_BR signaling. In this experiment, we did not apply baclofen to the cells, given that GABA can be also released by OPCs via VAMP2 (Zhang et al., 2021).

9. Supp Fig 11c,d: Were animals tested for different total amounts of time? It would seem the sniffing time amounts (F/UF) should add up to roughly the same amount of time per mouse, but they vary wildly.

In the revised manuscript, we have plotted the data in a different way (Fig. 6e-g and Supplementary Fig. 14c-j). Indeed, the total sniffing time can vary enormously between mice. Some of the animals ignored familiar or stranger mice throughout the complete session, while some were very actively exploring. Therefore, among the whole dataset, the sniffing time of each animal is rather scattered.

10. Line 171: "...including a smaller volume of vGAT (Fig. 2h) and less puncta (Fig. 2i) at the OPC surface in cKO mPFC, which can explain the attenuated firing rate of the interneurons in the adult." It isn't clear to me what the authors mean here.

We have completely revised the paragraph (Page 7-9).

11. Line 742: Behavior was only performed in male mice. A group containing only males is not representative. Including some female mice is advised to increase the representativeness and applicability of the behavior results in this manuscript.

For this revised version, we have now included 15 ctl and 9 cKO female mice and depict the results together with the male data in different color codes (Fig 6c-m, Supplementary Fig. 14c-j).

12. Line 921: I assume you mean 1×10^5 cells and the exponent got dropped.

Thank you for the correction. The change has been made in the revised manuscript.

13. Line 957: Is "ia" supposed to be "ie"?

This has been corrected in the revised manuscript.

Reviewer #2 (Remarks to the Author):

Authors analyzed the impact of conditional KO of GABA_BR from OPC in mice and demonstrated that such a manipulation impacts the differentiation of PFC OPC maturation, and also PFC PV interneuron myelination and their density (inhibiting apoptosis). They also found reduction of inhibitory inputs onto PFC pyramidal neurons and deficits in social and object novelty behavior tasks. The authors also demonstrated an in vitro evidence that PV apoptosis is regulated by OPC derived TWEAK signaling under the regulation of GABA_BR. Overall, I found the findings are interesting, and are consistent with the recent findings demonstrating a bi-directional link between PV interneuron and OPC through GABA signaling onto OPC (Benamer et al Nat Communications 2020). It adds new insights that not only GABA_AR but also GABA_BR plays a key role in this process, and also provide some novel mechanistic insights into how GABA signaling in OPC regulates PV apoptosis. However, to more convincingly support some of authors' claims, I suggest to more thoroughly characterize 1) GABA_BR deletion from OPC across development, 2) GABA_BR current changes in OPC, 3) myelination of excitatory neurons, 4) PV cell functional properties. Given that findings on TEWAK pathway is limited to in vitro setting, additional in vivo characterization and discussion on the mechanism will also strengthen the manuscript.

Major comments:

#1 Throughout the manuscript, conditional knock-down of OPC-GABA_BR needs to be better validated across development. It is important to show the timeline of deletion of GABA_BR for some of experiments when analysis is conducted at early ages such as p5-7, p10-14 (data similar to Fig1c and d at earlier age). Even if tdTomato is positive, it does not necessary mean GABA_BR per se is successfully knocked-out.

We thank the reviewer for the valuable comments. We do agree that tdTomato positive cells are not necessarily cKO OPCs, however, we estimate a similar recombination efficiency of reporter and the gabbr1 deletion in the cKO mPFC. As we have shown previously, the accessibility of floxed allele (i.e. STOP cassette of Rosa26-STOP^{f/f}-tdTomato (R26-tdT) and exon VII-VIII of gabbr1) are comparable in astrocytes (Jahn et al., 2018). Using the same R26-tdT and gabbr1 mice strains, we also observed a similar recombination efficiencies of tdTomato and gabbr1 deletion (Fig. 1c, d). The Western blot results showed about 64.2 % reduction of GABA_{B1} protein at 9w (Fig. 1d), when about 76 % cKO OPCs were recombined for tdT expression (% of Pa⁺tdT⁺/Pa⁺, Fig. 1c). Considering the cell purity of MACs OPCs

(85 %), we believe that the reporter recombination efficiency faithfully indicate the rate of gene deletion (64.2 % ($GABA_{B1}$ reduction)/85 % (OPC purity)=75.5 % (*gabbr1* deletion efficiency) vs 76 % (% of Pa^+tdT^+/Pa^+)). In the revised manuscript, we have added the reporter recombination efficiency of p5 (**Supplementary Fig. 12c**) and p10 groups (**Supplementary Fig. 7e, f**) and to indicate the potential gene deletion rate. At p5-7 and p10-14, about 40-60 % and 70-80 % OPCs were expressing tdTomato, respectively. Therefore, we expect similar extent of *gabbr1* deletion at corresponding time points.

#2 It would be helpful to show the extent of both GABA_A and GABA_B receptor mediated currents in developing OPC, and how knock-out of GABA_BR in OPC impacts these currents and membrane potential (given that recent publication of GABA_AR knock-out from OPC showed similar results, Benamer et al Nat Communications 2020).

*We are grateful for the reviewer's suggestion. To address whether there is compensatory machinery of GABA_AR in OPCs by deletion of GABA_BRs, we performed whole cell patch clamp recordings of OPCs at p11-14. GABA was bath-applied (in the presence of CNQX and DAP5 to block AMPAR and NMDAR currents, respectively) and evoked currents in OPCs were recorded. Since the current amplitudes of ctl and cKO OPCs were identical (**Supplementary Fig. 8e**), we excluded a compensatory response by GABA_ARs in the cKO OPCs. The application of picrotoxin and SR 95531 (noncompetitive and selective competitive GABA_AR antagonist) completely abolished any GABA evoked currents, therefore, we concluded that activation of GABA_BR in OPCs does not elicit any G protein activated K^+ current at this age. In conclusion, OPC-specific GABA_BR deletion does not affect GABA_AR function.*

#3 Selectivity of deficits in intracortical myelination of PV interneurons is not well supported by the presented data. It is more convincing if analysis is extended to myelinated axons of excitatory neurons within cortex (such as thalamo-cortical projections) to make such a claim.

*We do agree with the reviewer that the analysis on excitatory neuron myelination could strengthen our conclusion of an interneuron-specific hypomyelination. In the revised manuscript, we have analyzed the myelin volume of PV⁺ and total axons by performing MOG/PV/SMI 312 triple immunostaining in mPFC (**Fig. 2d, e**). Our results showed an unperturbed myelin volume of total axons while decreased myelin volume of PV⁺ axons in cKO mPFC (**Fig. 2e**). In addition, neither the myelin structures (i.e. node and paranode length) nor the MBP expression was altered in the cKO mice primary motor cortex (**Supplementary Fig. 4**). As well, our data from corpus callosum, where the axons are excitatory with their soma located in the cortex (Fame et al., 2011), demonstrated that the myelin structure (node,*

paranode length and g-ratio, **Supplementary Fig. 5**), myelin protein (MBP, **Fig. 1k**) expression as well as the action potential conduction velocity (**Supplementary Fig. 5i**) remained the same. Therefore, our data suggest that in the medial prefrontal cortex, the hypomyelination is rather selective to inhibitory neurons in OPC-GABA_BR cKO mice.

#4 line 157, 247-8, 289: Hypoactivity of PV interneurons not well supported. Only data indirectly related is reduced sIPSC frequency from pyramidal neurons. Direct electrophysiological characterization of PV cells (e.g. excitability) in slice, or in vivo activity (e.g. optogenetic tagged recording or early immediate gene expression in PV) will help to support this claim.

*We are grateful to the reviewer's suggestion of using an additional, independent marker of neuronal activity such as immediate early gene expression. To directly assess the PV interneuron activity, we performed cFos and PV double immunostaining at p10, p14 and 9w. At p10 PV⁺cFos⁺ cell density was yet unaffected at cKO mPFC (**Supplementary Fig. 8f**), although the vGAT density and its volume of immunolabel was already decreased (**Fig. 3i-k**). Then, at p14, both the density of PV⁺cFos⁺ interneurons and the ratio of PV⁺cFos⁺ of all PV⁺ interneurons were decreased by about 60 % in the cKO mPFC (**Supplementary Fig. 8g**). In addition, by assessing sPSCs of OPCs, we also observed that sPSC frequency was reduced by 64 % in the cKO mPFC at p14 (**Fig. 3m**), well in line with the cFos results. Application of carbachol, which activates interneurons, augmented the frequency in both ctl and cKO OPCs, however, the cKO sPSC frequency was still smaller than that of ctl (**Fig. 3m**). Therefore, these data together suggest that interneuron activity is declined in the cKO mPFC at p14.*

#5 In vivo evidence of the role of TWEAK is limited. I suggest to at least show the expression pattern of TWEAKR especially in PV cells in tissue by immunohistochemistry. As all associated data is from cultured condition, limitation of the findings (lack of in vivo study) also needs to be discussed in the discussion section.

*We thank the reviewer for this valuable suggestion. We have performed double immunostaining of TWEAKR and PV at p5 mPFC (**Fig. 5e**).*

#6 One of the limitations of the current study is the lack of mechanism of GABA_BR regulation of TWEAK secretion. Can author discuss this more?

We thank the reviewer for the suggestion and have added this point in the discussion.

Page 16-17, line 411: 'GABA_BRs are G-protein coupled receptors. In cultures of OPCs, activation of GABA_BR negatively regulates adenylyl cyclase and reduces cAMP levels⁴⁷. Subsequently, protein kinase A activity is suppressed, followed by impeded nuclear translocation of cAMP response element binding protein (CREB) affecting gene expression, e.g. the expression of brain derived neurotrophic factor (BDNF) or AMPA (α -amino-3-hydroxy-5-methyl-4-isoxazolepropionic acid) -type glutamate receptor GluA1 subunit^{48, 49}. A recent study suggested that activation of GABA_BR in cultured OPCs can also activate Akt/Src kinases required for OPC differentiation¹⁴. Additional studies are necessary to elucidate which class of G proteins (G_{ai} and/or G_o) transmit GABA_BR signaling in OPCs to elicit TWEAK expression and release.'

Minor comments

Is there any PV density difference in different layers of cortex in mPFC? (Fig2)

We have reanalyzed our data more specifically according to reviewer's suggestion. We analyzed the density of PV⁺ cells in three defined regions (layer I, layer II/III and layer V/VI). Our data show that the PV⁺ cell density is overall increased in the three selected regions of the cKO mPFC (**Fig. 2f, g**).

Fig4b: Is the power difference selective to gamma band? Please show data of other power band.

In the revised manuscript, we have also included the other power bands (**Supplementary Fig. 14a, b**). In short, the relative power of theta, alpha and sigma waves are increased as well in the cKO mouse brain. Theta and gamma bands are associated with cognition, particularly gamma band represents the activities of parvalbumin interneurons (Sohal et al., 2009). Alpha band is related to the attention and alertness (Fu et al., 2001), while sigma is associated with non-rapid eye movement sleep (Lecci et al., 2017).

Throughout the manuscript, it is unclear if the sample number is cell, section, or animal. Please provide the information explicitly.

We apologize for our unclear statistics. In the revised manuscript, we have indicated the meaning of 'n' and sample size in each figure legend.

line 256. My understanding is the new object recognition is not PFC dependent.

Prefrontal cortex (PFC) has been suggested to be involved in the episodic memory (Rugg et al., 2002; Hawco et al., 2013; Robin et al., 2015), which belongs to the recognition memory (Rugg and Yonelinas,

2003; Squire et al., 2004). The recognition memory can be defined as the memory that allows an individual to judge the prior occurrence of a particular stimulus or episode. The first attempts to analyze recognition memory in rodents' used reward-based tasks (Kesner et al., 1993), but this behavior test requires many training trials and animals are often food-deprived. To avoid these problems, the spontaneous object recognition task was developed (Ennaceur and Delacour, 1988; Leger et al., 2013). In addition, together with orbitofrontal cortex, PFC is indeed required for object recognition (Reid et al., 2014). Therefore, here we selected new object recognition, combining with social behavior tests, as paradigms to assess the functional change of mPFC in the OPC-GABA_BR cKO animals. Together with the social cognition study, the new object recognition results further substantiated cognitive impairment in the mutant mice.

line 275-6: As manipulation is not selective at PFC, link between PFC and behavior is not established by this study.

We do agree with the reviewer that the OPC-GABA_BR ablation is not restricted to the prefrontal cortex (PFC). However, social cognition or social interaction is preferentially related to the PFC. Lesion in the mPFC led to a severe social cognition impairment. In addition, the hypomyelination in the mPFC resulted in social cognition defects (Makinodan et al., 2012; Grossmann, 2013; Swire et al., 2019). All these studies suggest that mPFC contributes significantly to the social cognition. Therefore, we concluded that our data supports the notion that the aberrant myelination and activity of interneurons in the cKO mPFC contributes to impaired cognition.

It would be helpful to discuss similarities and differences between the action of GABA_AR and GABA_BR in OPCs to PV interneurons (especially findings reported in Benamer et al 2020).

We thank the reviewer for the suggestion. In the revised manuscript, we have included a respective paragraph to the discussion.

Page 17-18, line 441: 'Interestingly, also the early loss of the ionotropic GABA_AR γ 2 subunit in OPCs reduced the firing rate of presynaptic PV⁺ interneurons as well as a similar myelin defect⁴. Obviously, the very different signal pathways of ionotropic and metabotropic GABA receptors merge and similarly affect PV interneuron activity, OPC-axon contacts and myelin gene expression. However, the impact on myelin structures was rather distinct. GABA_AR deletion resulted in prolonged lengths of nodes and internodes with a decreased density of paranodes⁴, while only an extended length of paranodes was detected in the OPC-GABA_BR cKO mPFC of our study. In addition, GABA_ARs facilitated interneuron maturation in the

juvenile (p24) somatosensory cortex, while GABA_BRs regulate the elimination of interneurons during the first two postnatal weeks. Apparently, both receptors optimize the density and activity of PV interneurons, cooperating jointly but employing different machineries at distinct time points.'

Reviewer #3 (Remarks to the Author):

In the manuscript “Impaired bidirectional communication between interneurons and oligodendrocyte precursor cells affects cognitive behavior”, Fang et al present exciting findings providing new insights on the roles of early communication between interneurons and OPCs in mPFC myelination, interneuron activity, cortical network function and cognitive behavior. Specifically, the authors show that interruption of GABAB receptor-mediated signaling to OPCs (via deletion of *gabbr1* subunit of GABABRs selectively in OPCs) at the end of the first postnatal week results in impaired mPFC myelination and reduced inhibitory network activity, leading to deficits in cognition and social behavior. Besides they revealed that the release of the TNF-related cytokine TWEAK by OPCs, in a GABABR mediated manner, induces neuronal apoptosis to regulate the density of parvalbumin (PV) interneurons.

Therefore, the study of Fang et al. sheds new light on the role of bidirectional communication between interneurons and OPCs in the regulation of cell densities and myelination. The study combines various approaches, from patch-clamp recordings to testing behavior of animals that makes the conclusions solid and of significant interest to the broad fields of activity-dependent myelination and cortical inhibitory circuits. Overall, the study significantly contributes to understand the role of interneuron-OPC interactions. However, some conclusions are confusing and some clarifications will be helpful:

1. In Figure 1, the authors observed a decrease of mature oligodendrocytes (OL) associated to a hypomyelination in the mPFC of the KO compared to the Control while they found no differences in corpus callosum. Did they also observe these effects in other cortical area, or is it specific of the mPFC?

In the primary motor cortex, we have also observed the decrease of oligodendrocyte density at 2w and 9w age (Supplementary Fig. 2). However, neither the mean length of paranode or node, nor the MBP expression was perturbed (Supplementary Fig. 4h-p). Therefore, we hypothesize that in different brain regions, the OPC-GABA_BRs may function differently.

2. The authors observed a decrease in mature OL in the mPCF KO without affecting the density of OPCs and conclude that OPC-GABABR play a role in OPC differentiation. However, the fact that the pool of OPCs is not affected may suggest that there is an OL cell death increase that causes the difference in OL cell number rather than less differentiation. This point needs more explanation by looking at Caspase-3 expression in OL, for instance.

We agree with the reviewer that oligodendrocyte apoptosis needs to be considered. In the revised manuscript, we have performed double immunostaining of CC1 with cleaved caspase-3 (CC-3) at p14, at this time point we detected about 66 % reduction of the OL density. Overall, CC-3⁺ oligodendrocytes were quite rare at p14 and we did not observe differences of CC1⁺CC-3⁺ oligodendrocyte numbers in *ctl* and *cKO* mPFC (**Supplementary Fig. 9**). Therefore, our data strongly suggest that it is the OPC differentiation rate that is mitigated in the *cKO* mPFC rather than a change of oligodendrocyte survival.

3. Besides, they observed in the KO mPFC an increase in the paranodal length without affecting the node length. In Figure 1j, the authors show differences in the distribution of the paranodes length, but is there also a significant difference of their mean size.

We apologize for the confusing organization of our previous figures. The mean size of the paranode length is also increased in the *cKO* mPFC (**Supplementary Fig. 4b**). The putative impact and the functional readout of longer paranodes, we have also addressed in the “Discussion” of the revised manuscript.

The authors should also discuss more how this could affect PV-interneuron function and properties. For instance, could longer paranode affect protein clustering and potassium channel expression (Kv) in the juxtaparanode region?

We thank the reviewer for this valuable suggestion. The elongation of paranode will be accompanied with redistribution of Kv1 channels from under juxtaparanode towards paranode in the brain of multiple sclerosis and aging (Arancibia-Carcamo and Attwell, 2014). These exposed Kv1 channels are more active. As a result, these changes hinder the action potential propagation. We have implemented this interpretation in the ‘Discussion’ section.

Page 18-19, line 466: ‘The elongation of paranode will be accompanied with redistribution of Kv1 channels from under juxtaparanode towards paranode in the brain of multiple sclerosis and aging⁵⁶. These exposed Kv1 channels are more active. As a result, these changes hinder the action potential propagation, which in turn reduces the synaptic communication.’

4. In Figure 1m-n, the authors report in the KO a decrease in the volume of myelin sheaths wrapping PV interneurons by doing only a PV and MBP double immunostaining. However, in figure 1m showing 3D reconstructions of myelinated PV+ axons, we see less PV+ processes in KO compared to Control mice. The reported differences could thus be due to less PV+ axons rather than less

myelinated PV+ axons. Did the authors normalize the quantification of myelinated PV+ axons to the volume of PV+ axons? Could the authors give more details about how the quantification was done?

To strengthen the conclusion that GABABR of OPCs are involved specifically in myelination of inhibitory axons, the authors should stain specifically PV+ axons by using an axonal marker (SMI-312 for instance) in addition to PV and MBP double immunostaining, and normalized to the total volume of PV+ axons. Alternatively, they can quantify the co-localization of PV and MBP specifically in biocytin loaded PV-interneurons.

We apologize for the confusing description. In the revised version, according to the reviewer's suggestion, we now performed a triple immunolabeling MOG/PV/SMI 312 (pan axonal marker) to evaluate the PV interneuron myelination, and used the PV axonal volume for normalization. Our results exhibited that PV axonal myelination is significantly reduced in the cKO mPFC, while no differences were detected for the myelination of all axons (Fig. 2e). This result is consistent with the observation that MOG mRNA expression is not altered in the cKO mPFC (Supplementary Fig. 4g).

5. In Figure 2b, the authors show an increase of PV interneuron density in the KO mPFC. What about the density of PV-interneurons among layers? Do they observe these differences if they compare the densities between upper and deeper layers, specifically? This is relevant in this study since they studied neuronal activity in layer 5.

We have re-analyzed the density of PV⁺ cells in the mPFC. After selecting three regions, including layer I, layer II/III and layer V/VI, we found an increased interneuron density in the three selected areas of the cKO mPFC (Fig. 2f, g). The surplus of PV interneurons in the mPFC appears to be a more general and layer-independent phenotype in the cKO mice.

6. The authors quantified synaptic vGAT density of OPC surface and observed a decrease in the density of vesicles on OPCs but also on other cells in the KO. First, this effect is not specific of OPCs and the authors should discuss how a loss of GABABR signaling to OPCs could generate a global decrease in puncta densities. Besides, the conclusion that less GABAergic vesicles on OPC surface could explain the attenuation of firing rate in interneurons (line 171) based on those results is not correct and should be modified. What is the rationale for quantifying the density of vesicular GABA transporter close to OPCs as a readout of interneuron activity? To correlate the decrease of vesicle densities and neuronal GABAergic activity, the authors should quantify miniature inhibitory postsynaptic currents in neurons.

We thank for the reviewer's comments and apologize for the unclear statement. We performed vGAT immunostaining to assess the interneuron activity. We have analyzed the total vGAT density and volume as a qualitative readout of overall interneuron activity (Fig. 3i, j). Both density and volume were decreased by 75 % in the cKO mPFC, suggesting hypoactivity of interneurons. In parallel, we also performed PV and cFos double immunostaining to assess PV neuron activity at p10 and p14 (Supplementary Fig. 8f, g). Quantification of the data showed that in the cKO mPFC, PV interneurons expressed less cFos, an immediate early gene which has been correlated with neuronal firing rates. In summary, these data suggest that PV interneuron activity is mitigated in the cKO mPFC during development.

The authors also conclude from these results that interneurons exhibit lower activity and transmit less GABAergic signals to OPCs. Again, this assumption is not correct and the author should quantify directly GABAergic activity in OPCs by patch-clamp recordings to reach this conclusion.

To evaluate the inhibitory inputs on OPCs in the cKO mPFC during development, we recorded sPSC on OPCs at p11-14 (Fig. 3l-n). Our results showed that the frequency of sPSC on OPCs was reduced by 60 % in the cKO mice while the current amplitude remained unaffected, suggesting OPCs receive less inhibitory input. In addition, the application of carbachol, which activates muscarinic receptors (mainly on neurons, (Lin and Bergles, 2004)), also increased firing rates in ctl and cKO brains. However, the frequency of the cKO cells approached only 60 % of that of ctl cells. Therefore, our data strongly suggest that OPCs receive less GABAergic input in the OPC-GABA_BR cKO mPFC.

References:

- Arancibia-Carcamo IL, Attwell D (2014) The node of Ranvier in CNS pathology. *Acta Neuropathol* 128:161-175.
- Cheadle L, Rivera SA, Phelps JS, Ennis KA, Stevens B, Burkly LC, Lee WA, Greenberg ME (2020) Sensory Experience Engages Microglia to Shape Neural Connectivity through a Non-Phagocytic Mechanism. *Neuron* 108:451-468.e459.
- Crosby KC, Gookin SE, Garcia JD, Hahm KM, Dell'Acqua ML, Smith KR (2019) Nanoscale Subsynaptic Domains Underlie the Organization of the Inhibitory Synapse. *Cell Rep* 26:3284-3297.e3283.
- Ennaceur A, Delacour J (1988) A new one-trial test for neurobiological studies of memory in rats. 1: Behavioral data. *Behav Brain Res* 31:47-59.
- Fame RM, MacDonald JL, Macklis JD (2011) Development, specification, and diversity of callosal projection neurons. *Trends Neurosci* 34:41-50.
- Fu KM, Foxe JJ, Murray MM, Higgins BA, Javitt DC, Schroeder CE (2001) Attention-dependent suppression of distracter visual input can be cross-modally cued as indexed by anticipatory parieto-occipital alpha-band oscillations. *Brain Res Cogn Brain Res* 12:145-152.

- Grossmann T (2013) The role of medial prefrontal cortex in early social cognition. *Front Hum Neurosci* 7:340.
- Hawco C, Berlim MT, Lepage M (2013) The dorsolateral prefrontal cortex plays a role in self-initiated elaborative cognitive processing during episodic memory encoding: rTMS evidence. *PLoS One* 8:e73789.
- Jahn HM, Kasakow CV, Helfer A, Michely J, Verkhatsky A, Maurer HH, Scheller A, Kirchhoff F (2018) Refined protocols of tamoxifen injection for inducible DNA recombination in mouse astroglia. *Sci Rep* 8:5913.
- Kesner RP, Bolland BL, Dakis M (1993) Memory for spatial locations, motor responses, and objects: triple dissociation among the hippocampus, caudate nucleus, and extrastriate visual cortex. *Exp Brain Res* 93:462-470.
- Kessarlis N, Fogarty M, Iannarelli P, Grist M, Wegner M, Richardson WD (2006) Competing waves of oligodendrocytes in the forebrain and postnatal elimination of an embryonic lineage. *Nat Neurosci* 9:173-179.
- Lecci S, Fernandez LM, Weber FD, Cardis R, Chatton JY, Born J, Lüthi A (2017) Coordinated infraslow neural and cardiac oscillations mark fragility and offline periods in mammalian sleep. *Sci Adv* 3:e1602026.
- Leger M, Quiedeville A, Bouet V, Haelewyn B, Boulouard M, Schumann-Bard P, Freret T (2013) Object recognition test in mice. *Nat Protoc* 8:2531-2537.
- Lin SC, Bergles DE (2004) Synaptic signaling between GABAergic interneurons and oligodendrocyte precursor cells in the hippocampus. *Nat Neurosci* 7:24-32.
- Makinodan M, Rosen KM, Ito S, Corfas G (2012) A critical period for social experience-dependent oligodendrocyte maturation and myelination. *Science* 337:1357-1360.
- Reid JM, Jacklin DL, Winters BD (2014) Delineating prefrontal cortex region contributions to crossmodal object recognition in rats. *Cereb Cortex* 24:2108-2119.
- Robin J, Hirshhorn M, Rosenbaum RS, Winocur G, Moscovitch M, Grady CL (2015) Functional connectivity of hippocampal and prefrontal networks during episodic and spatial memory based on real-world environments. *Hippocampus* 25:81-93.
- Rugg MD, Yonelinas AP (2003) Human recognition memory: a cognitive neuroscience perspective. *Trends Cogn Sci* 7:313-319.
- Rugg MD, Otten LJ, Henson RN (2002) The neural basis of episodic memory: evidence from functional neuroimaging. *Philos Trans R Soc Lond B Biol Sci* 357:1097-1110.
- Sohal VS, Zhang F, Yizhar O, Deisseroth K (2009) Parvalbumin neurons and gamma rhythms enhance cortical circuit performance. *Nature* 459:698-702.
- Southwell DG, Paredes MF, Galvao RP, Jones DL, Froemke RC, Sebe JY, Alfaro-Cervello C, Tang Y, Garcia-Verdugo JM, Rubenstein JL, Baraban SC, Alvarez-Buylla A (2012) Intrinsically determined cell death of developing cortical interneurons. *Nature* 491:109-113.
- Squire LR, Stark CE, Clark RE (2004) The medial temporal lobe. *Annu Rev Neurosci* 27:279-306.
- Swire M, Kotelevtsev Y, Webb DJ, Lyons DA, French-Constant C (2019) Endothelin signalling mediates experience-dependent myelination in the CNS. *Elife* 8.
- Zhang X, Liu Y, Hong X, Li X, Meshul CK, Moore C, Yang Y, Han Y, Li WG, Qi X, Lou H, Duan S, Xu TL, Tong X (2021) NG2 glia-derived GABA release tunes inhibitory synapses and contributes to stress-induced anxiety. *Nat Commun* 12:5740.
- Zhang Y, Chen K, Sloan SA, Bennett ML, Scholze AR, O'Keefe S, Phatnani HP, Guarnieri P, Caneda C, Ruderisch N, Deng S, Liddelow SA, Zhang C, Daneman R, Maniatis T, Barres BA, Wu JQ (2014) An RNA-Sequencing Transcriptome and Splicing Database of Glia, Neurons, and Vascular Cells of the Cerebral Cortex. *J Neurosci* 34:11929-11947.

REVIEWERS' COMMENTS

Reviewer #2 (Remarks to the Author):

Revised manuscripts addressed most of the concerns I had to the initial manuscript. My only remaining minor issue is the lack of acknowledgement of the limitation of this study on not directly casually linking mPFC deficits to behavioral deficits. While the findings are consistent with the possible role of PFC in behavior changes, none of manipulations was conducted selectively at PFC. This limitation should be explicitly mention in the manuscript.

Reviewer #3 (Remarks to the Author):

In the revised version of their manuscript, the authors have addressed my previous comments and I have no more comments or concerns. Indeed the authors have addressed the comments of the reviewers by performing additional experiments, including new pieces of data providing new insights on the roles of early communication between interneurons and OPCs in brain function during development. They also re-arranged the text of the manuscript, and improved the Discussion section. I think that the manuscript has become clearer and more complete now, and appears stronger than the original version.

Point-to-point responses

Reviewer #2 (Remarks to the Author):

Revised manuscripts addressed most of the concerns I had to the initial manuscript. My only remaining minor issue is the lack of acknowledgement of the limitation of this study on not directly casually linking mPFC deficits to behavioral deficits. While the findings are consistent with the possible role of PFC in behavior changes, none of manipulations was conducted selectively at PFC. This limitation should be explicitly mention in the manuscript.

We are grateful for the reviewer's positive comments. We have implemented the limitation of the PFC selective manipulation in our study to the revised manuscript in the 'Discussion' session.

Discussion, line 489-491: 'Nota bene, since the genetic manipulation is not selective to the PFC, we cannot rule out contributions of other brain regions to the behavioral phenotype, which has to be left for future studies.'

Reviewer #3 (Remarks to the Author):

In the revised version of their manuscript, the authors have addressed my previous comments and I have no more comments or concerns. Indeed the authors have addressed the comments of the reviewers by performing additional experiments, including new pieces of data providing new insights on the roles of early communication between interneurons and OPCs in brain function during development. They also re-arranged the text of the manuscript, and improved the Discussion section. I think that the manuscript has become clearer and more complete now, and appears stronger than the original version.

We thank the reviewer for her/his positive comments.